©c Author(s) 2023. CC BY 4.0 License.





# Long-term aerosol particle depolarization ratio measurements with Halo Doppler lidar

Viet Le[1], Hannah Lobo[1], Ewan J. O'Connor[1], Ville Vakkari[1,2]

[1]Finnish Meteorological Institute, Helsinki, 00101, Finland

[2]Atmospheric Chemistry Research Group, Chemical Resource Beneficiation, North-West University, Potchefstroom, 2520, South Africa

*Correspondence to*: Viet Le (viet.le@fmi.fi)

**Abstract.**  It has been demonstrated that Halo Doppler lidars have the capability for retrieving the aerosol particle linear depolarization ratio at a wavelength of 1565 nm. However, the retrieval depends on an accurate representation of the

instrumental noise floor and the performance of the internal polarizer, whose stability have not been assessed in long-term operation. Here, we use four years of measurements at four sites in Finland to investigate the long-term performance of Halo Doppler lidars for aerosol particle depolarization ratio retrieval. The instrumental noise level, represented by noise-only signals in aerosol- and hydrometeor- free regions, shows stable performance for most instruments, but clear differences between individual instruments. For all instruments, the polarizer bleed-through evaluated at liquid cloud base remains reasonably

constant at approximately 1% with a standard deviation less than 1%. We find these results sufficient for long-term aerosol particle linear depolarization ratio measurements and proceed to analyse the seasonal and diurnal cycles of the aerosol particle depolarization ratio in different environments in Finland including in the Baltic Sea archipelago, boreal forest and rural sub-arctic. To do so, we further develop the background correction method and construct an algorithm to distinguish aerosol particles from hydrometeors. The four-year averaged aerosol particle depolarization ratio ranges from 0.07 in sub-arctic

Sodankylä to 0.13 in the boreal forest in Hyytiälä. At all sites, the aerosol particle depolarization ratio is found to peak during spring and early summer, even exceeding 0.20 at the monthly-mean level, which we attribute to a substantial contribution from pollen. Overall, our observations support the long-term usage of Halo Doppler lidar depolarization ratio including detection of aerosols that may pose a safety risk for aviation.

## 1 Introduction

Information on the aerosol vertical distribution in the atmosphere is vital for many applications. For instance, the direct radiative effects of aerosols can be quite different if the aerosol layer is situated above a cloud layer rather than within the boundary layer, while aerosol indirect radiative effects occur only if aerosols are immersed within the cloud (IPCC, 2021). The impact of aerosol on clouds and the radiative balance of the Earth are among the largest uncertainties in our understanding (IPCC, 2021). Air quality and associated adverse health effects (Di et al., 2017) are determined by surface concentrations,



which, however, are strongly affected by the vertical structure of the boundary layer (Kanawade et al., 2020; Wang et al., 2020). Finally, from an aviation point of view, high resolution profiles of aerosol vertical distribution could play a crucial role in mitigating the impact of hazardous aerosol emissions (Hirtl et al., 2020).

Aerosol vertical profiles can be observed with a number of different methods, such as in-situ instruments mounted on different platforms including research and commercial aircraft (Johnson et al., 2008; Pratt and Prather, 2010), tethered balloons

(Creamean et al., 2021; Rankin and Wolff, 2002; Hara et al., 2013), hot air balloons (Petäjä et al., 2012), zeppelin (Rosati et al., 2016) and unmanned aerial vehicles (Brus et al., 2021; Mamali et al., 2018). Comprehensive aerosol properties can be obtained from in-situ measurements: mass, size distribution, chemical composition; but the capabilities of the chosen platform limit the temporal resolution at which profiles can be obtained and the vertical extent of the profiling. On the other hand, active remote sensing with lidar only retrieves the optical properties of aerosol but is capable of continuous observations of the

vertical structure of the atmosphere. Space-borne lidars such as CALIPSO (Cloud-Aerosol Lidar and Infrared Pathfinder Satellite Observation; Winker et al., 2009) cover the globe but with low temporal and spatial resolution due to their very narrow swath (Baars et al., 2017). Airborne lidars, such as HSRL-1 (High Spectral Resolution Lidar) built and operated by NASA Langley Research Center (Hair et al., 2008), provide good spatial coverage but are costly and only able to operate for a relatively short period at a time. A combination of temporal and spatial coverage can be achieved through ground-based

networks of lidars, such as EARLINET (European Aerosol Research Lidar Network; Pappalardo et al., 2014) and Finland's ground-based remote-sensing network (Hirsikko et al., 2014). These lidar networks enable the monitoring in real-time of vertical profiles of aerosol in different environments, and consequently facilitate the detection of elevated aerosol layers and the investigation of vertical atmospheric properties throughout the seasons.

In active remote sensing, one of the most important parameters for characterising aerosol is the aerosol particle depolarization

ratio ($\delta_{aerosol}$), which is the ratio of the co-polar and cross-polar signals backscattered from aerosol. This parameter is used to distinguish between spherical and non-spherical particles (Burton et al., 2012; Baars et al., 2017; Mamouri and Ansmann, 2016) and is therefore essential in differentiating aerosol types (Illingworth et al., 2015) such as smoke, dust, marine and ash. Typically, $\delta_{aerosol}$ is measured using Raman lidar (Engelmann et al., 2016; Baars et al., 2016) or micropulse lidar (Flynn, et al., 2007; Córdoba-Jabonero et al., 2018) at wavelengths such as 355 nm, 532 nm, 710 nm or 1064 nm. This study is conducted

using data from Halo Doppler lidars at 1565 nm, which is a relatively new addition to the suite of wavelengths used for $\delta_{aerosol}$ retrieval (Vakkari et al., 2021).

Halo Doppler lidars (Pearson et al., 2009) are the core remote sensing instruments in the Finnish ground-based remote sensing network. In this study, we analysed data from these instruments at four different locations in the network from 2016 to 2019. Liquid clouds were identified and the depolarisation ratio, δ, at liquid cloud base collected to derive the depolarisation bleed

through (Vakkari et al., 2021) and its temporal evolution for five different Halo Doppler lidar instruments. In addition, the stability of every Halo Doppler lidar in the network was also assessed through the time series of the signal-to-noise ratio in aerosol- and hydrometeor- free regions. Furthermore, an aerosol identification algorithm was created to enable the separation





of aerosol and hydrometeors similar to the Cloudnet algorithm (Illingworth et al., 2007; Tukiainen et al., 2020), but based only

on one instrument. The algorithm was used to extract $\delta_{aerosol}$ at all locations and subsequently, the overall statistics of $\delta_{aerosol}$ at 1565 nm in Finland across four years. These statistics can improve aviation safety by providing a baseline of $\delta_{aerosol}$ at this wavelength, so that potentially hazardous layers such as smoke and volcanic ash can be separated from the natural aerosol more easily.

## 2 Materials and methods

### 2.1 Halo Doppler lidar

The Finnish remote sensing network deploys Halo Doppler lidars in several measurement stations across Finland (Hirsikko et al., 2014). This study uses data from Utö, Hyytiälä, Vehmasmäki and Sodankylä (Figure 1). Each location has a different environment, enabling comparisons between both urban and rural; marine, continental, and sub-arctic regions. Detailed descriptions and the study period for each location are shown in Table 1.

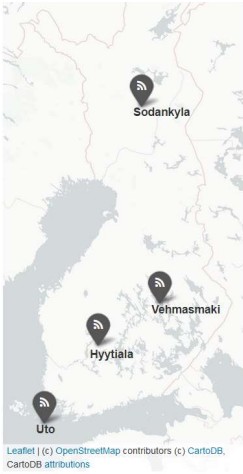

**Figure 1: Locations of the instruments across Finland**

| Site | Description | Instrument | Instrument ID | Study period |
|------|-------------|------------|---------------|--------------|
| Utö 59.77ºN,21.37ºE | Island | StreamLine | Utö-32 | 1.1.2016 - 16.8.2017 |
| | | StreamLine XR | Utö-32XR | 22.11.2017 - 31.12.2019 |
| Vehmasmäki - Kuopio 62.89ºN,27.63ºE | Semi-urban/rural | StreamLine Pro | Vehmasmäki-53 | 1.1.2016 - 31.12.2019 |



| Hyytiälä (SMEAR II) 61.84°N,24.29°E | Rural (boreal forest) | StreamLine StreamLine | Hyytiälä-33 Hyytiälä-46 | 1.1.2016 -7.8.2017 9.10.2017 - 31.12.2019 |
|---|---|---|---|---|
| Sodankylä 67.37°N,26.62°E | Arctic rural | StreamLine Pro | Sodankylä-54 | 19.6.2017 - 20.11.2019 |

**Table 1: Description of instrument locations**

The Halo Photonics StreamLine Doppler lidars (Halo Doppler lidars) operated by the Finnish Meteorological Institute are 1565 nm pulsed Doppler lidars transmitting linearly polarized light and equipped with heterodyne detectors that can switch

between recording the return in two channels parallel and orthogonal with respect to the transmitted polarisation (Pearson et al., 2009), termed co-polar (parallel) and cross-polar (orthogonal). These lidars are fibre optic systems, utilizing solid-state lasers, and capable of operating continuously for a long period of time (Harvey et al., 2013). In addition, these systems conform to eye-safety requirements as they operate at high-pulse repetition and low-pulse energy mode with a 1565 nm laser (Pearson et al., 2009).

Three versions of Halo Doppler lidars are utilized in this study: StreamLine, StreamLine Pro and StreamLine XR lidars (HALO PHOTONICS | StreamLine series - Product, 2021). The StreamLine and StreamLine XR lidars are capable of full hemispheric scanning. Designed for harsher environments, the StreamLine Pro lidar has no external moving parts, limiting the scanning to within a 20-degree cone around Zenith. The StreamLine XR has higher power and a lower pulse repetition frequency, thus can observe up to 12 km in range above ground level (a.g.l) compared to only 9.6 km for the StreamLine and Streamline Pro. Key

specifications of all instruments are shown in Table 2.

The operational mode of each instrument varies with location. The standard operation mode consists of continuous vertical staring with periodical switching to velocity-azimuth display scans for obtaining vertical profiles of the horizontal wind. For about 10-30 seconds in every hour the instruments perform a periodical background noise determination. Only data from the vertical staring mode is utilized in this study, so occasional gaps in data availability are due to lower elevation angle scanning

and the background noise determination.

In vertical staring mode, the instrument emits pulses of polarized light into the atmosphere and then records the returned vertical Doppler velocity (w) and signal-to-noise ratio in both co- and cross-polar ($SNR_{co}$ and $SNR_{cross}$) sequentially. From these measurements, profiles of attenuated backscatter ($\beta'$) and particle depolarization ratio ($\delta$) are derived.

In Halo Doppler lidars, measurements of $SNR_{co}$ and $SNR_{cross}$ are taken sequentially. For example, if the integration time of the

instrument is set to 7 seconds, then $SNR_{co}$ is collected for 7 seconds and then $SNR_{cross}$ is collected during the next 7 seconds (Vakkari et al., 2021), and the resulting $\delta$ will be presented with 14 seconds time resolution. Such long-time resolution can cause issues especially for cloud measurements, if e.g., cloud base height changes between co- and cross-polar measurements. However, aerosol measurements in low signal conditions require extended integration time to reduce noise. Thus, integration time is always a compromise between high time resolution and low background noise; and care must be taken to ensure that $\delta$



is calculated from the same part of the cloud. On the other hand, aerosol is expected to be well-mixed within each aerosol layer so $\delta_{aerosol}$ can be calculated from 1-hour averaged measurements to minimize noise effect on weak signals.

Additionally, data from a co-located Vaisala CL31 ceilometer at Utö was utilised to determine effective beam diameter and focal length according to (Pentikäinen et al., 2020) for the Utö-32XR instrument. For the non-XR instruments, attenuated backscatter was determined from the post-processed SNR using the 2 km focal length set in the firmware (Table 2).

| Specification | Values |
|---|---|
| Wavelength | 1565 nm |
| Beam divergence | 3.3e-05 rad |
| Laser Energy | 1e-05 J |
| Lens diameter | 0.06 m |
| Number of samples per range gate | 10 |
| Range resolution | 30 m |
| Pulse length | 200 ns |
| Minimum range | 90 m |
| Pulse repetition frequency | 10 kHz (Utö-32XR) |
| | 15 kHz (Other instruments) |
| Focus | Infinity (Utö-32XR) |
| | 2 km (Other instruments) |
| Integration time | 30 s (2016-01-01 to 2016-07-12, Utö-32) |
| | 5 s (2016-07-13 to 2017-11-21, Utö-32) |
| | 2 s (Utö-32XR) |
| | 6 s (Hyytiälä-33, Hyytiälä-46, Vehmasmäki-53, Sodankylä-54) |
| Bandwidth | 25000 MHz (Vehmasmäki-53, Sodankylä-54) |
| | 50000 MHz (Other instruments) |
| Nyquist velocity | 9.7 m s$^{-1}$ (Vehmasmäki-53, Sodankylä-54) |
| | 19.4 m s$^{-1}$ (Other instruments) |

**Table 2: Instrument configurations**





### 2.1.1 Instrumental noise floor

Typically, every hour, the instrument performs a background check to determine the range-resolved background noise level, which is then used in the firmware to calculate the SNR. The data in this study have been post-processed with the background correction algorithm as described by Vakkari et al., (2019), which removes the bias in SNR, i.e., SNR is centred on 0 when there is no signal. This algorithm enables the evaluation of the temporal evolution of the SNR in the aerosol- and hydrometeor-free zone, which can be used to assess the long-term changes of the noise floor for each instrument in the network. Here, the aerosol- and hydrometeor- free part of the SNR profiles have been manually collected for all the instruments throughout the whole study period.

### 2.1.2 Instrumental internal polarizer

In Halo Doppler lidars, an internal polarizer is used to measure the co- and cross-polar signals. The design of the Halo Doppler lidars does not facilitate user calibration of the polarizer performance, unlike aerosol research lidars such as PollyXT (Baars et al., 2011). Therefore, $\delta$ at liquid cloud base, which is defined in this study as

$$\delta = \frac{SNR_{cross}}{SNR_{co}}, \tag{1}$$

is used to estimate the internal polarizer performance, or bleed-through (Vakkari et al., 2021).

Single scattering from a spherical droplet does not change the incident polarization state into the 180-degree backward direction (Liou and Schotland, 1971), which results in no return signal at cross polarization. Hence, it is expected that $\delta$ at pure liquid cloud base is zero. However, as the laser beam penetrates further into the cloud, the observed $\delta$ gradually increases as the multiple-scattering contribution increases (Liou and Schotland, 1971; Hu et al., 2006), which is demonstrated in figure 2.

In order to determine the long-term performance of the internal polarizer, statistics of $\delta$ at liquid cloud base need to be obtained. Previous studies have introduced multiple algorithms based on attenuated backscatter profile to detect liquid cloud base (Zhao et al., 2014; Tuononen et al., 2019). In this study, taking the additional advantage of Halo Doppler lidar's capability in observing $\delta$ and w, we develop a new approach based on Tuononen et al., (2019) to detect the liquid cloud base.

First, a period of a liquid cloud layer is chosen through visual inspections. Within this period, the following criteria are used to choose suitable profiles to determine $\delta$ at liquid cloud base:

1. $\beta' > 10^{-5}$ m$^{-1}$ sr$^{-1}$,
2. $\delta$ increases monotonously from the lowest range gate (cloud base) where $\beta'$ exceeds $10^{-5}$ m$^{-1}$ sr$^{-1}$ to the range gate of maximum SNR$_{co}$,
3. vertical extent from the lowest range gate where $\beta'$ exceeds $10^{-5}$ m$^{-1}$ sr$^{-1}$ to the range gate of maximum SNR$_{co}$ does not exceed 100 m, and
4. w is between -0.5 m s$^{-1}$ and 0.5 m s$^{-1}$ for the range gates where $\beta' > 10^{-5}$ m$^{-1}$ sr$^{-1}$.

Criterion 2 reflects the increasing multiple scattering contribution (Liou and Schotland, 1971; Hu et al., 2006). Criterion 3 reflects the rapid attenuation of the signal inside liquid cloud (Tuononen et al., 2019; O'Connor et al., 2004) and reduces the





likelihood of including ice clouds in the analysis. Criterion 4 removes profiles that may contain precipitation and to ensure that the observed cloud base does not fluctuate in height too much, i.e., $SNR_{co}$ and $SNR_{cross}$ signals observe the same part of the cloud.

Values of δ at liquid cloud base were collected from each site throughout the whole study period. Where possible, at least one liquid cloud case per week was selected. The mean and standard deviation of δ at cloud base were then calculated and used to determine the bleed through for each instrument and to investigate its stability over time.

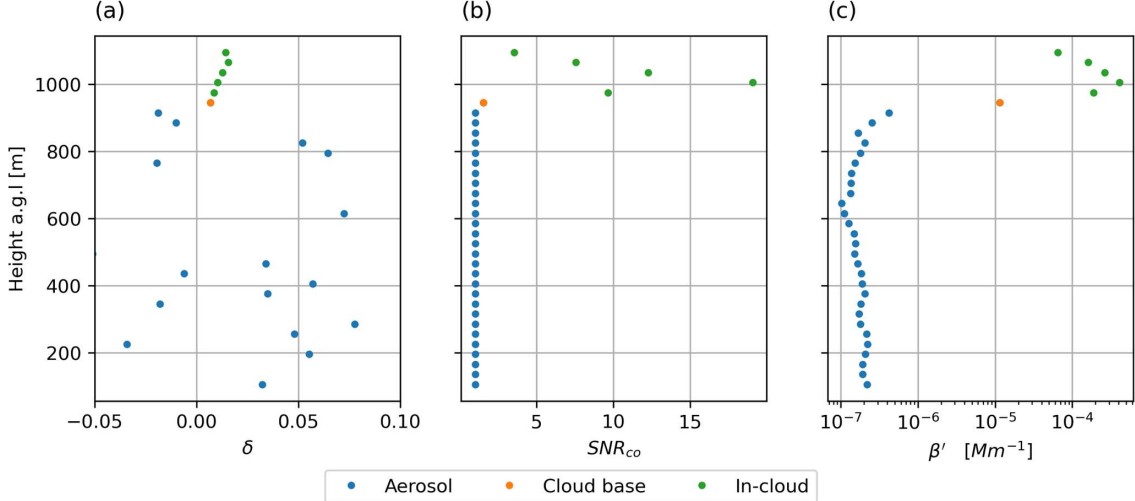

**Figure 2: Atmospheric profiles observed in 2018-09-24 by Hyytiälä-46 up to 1.4 km a.g.l. a) Depolarization ratio (δ), b) Signal-to-noise ratio in the co-polar channel ($SNR_{co}$) c) Attenuated backscatter (β')**

The signal from liquid cloud droplets is very strong, as seen also in Figure 2b. Hence, there is a risk of signal saturation. Given the low bleed through we would expect $SNR_{co}$ to get saturated before $SNR_{cross}$, which would appear as increasing δ in such cases. Figure S1 displays the 2D histogram of $SNR_{co}$ and $SNR_{cross}$ observed at liquid cloud base in Utö with the XR instrument. $SNR_{co}$ and $SNR_{cross}$ follow a linear relationship with the gradient corresponding to the determined bleed-through, indicating that saturation is not an issue, except maybe for some scattered points where $SNR_{co} > 6$.



### 2.2 Aerosol identification algorithm

In order to study the seasonal pattern of $\delta_{aerosol}$, it is essential to distinguish aerosol from clouds and precipitation. For sites with a co-located cloud radar, one option would be to utilize the Cloudnet classification (Tukiainen et al., 2020; Illingworth et al., 2007), but since not all sites have full Cloudnet instrumentation (cloud radar, ceilometer, microwave radiometer), an algorithm using only Doppler lidar is required.

Using a $\beta$' threshold alone is not sufficient for separating aerosol and precipitation. Likewise, using a simple threshold on w alone is not sufficient for differentiating aerosol from light snowfall or precipitation from strong downdrafts. Hence, an algorithm utilizing $SNR_{co}$, $\beta$' and w in both the time and height domain was developed for distinguishing aerosol from larger hydrometeors. The aerosol identification (AI) algorithm developed here utilizes 2D-kernel manipulation to extract various features from the data and to determine the correct class for each data point. The resulting classes are background signal, aerosol, hydrometeor, and undefined. For this algorithm, hydrometeors are defined as cloud (liquid or ice) or precipitation (rain or snow) and do not include aerosol.

The detailed steps of the algorithm are shown in Figure S3 and explained below:



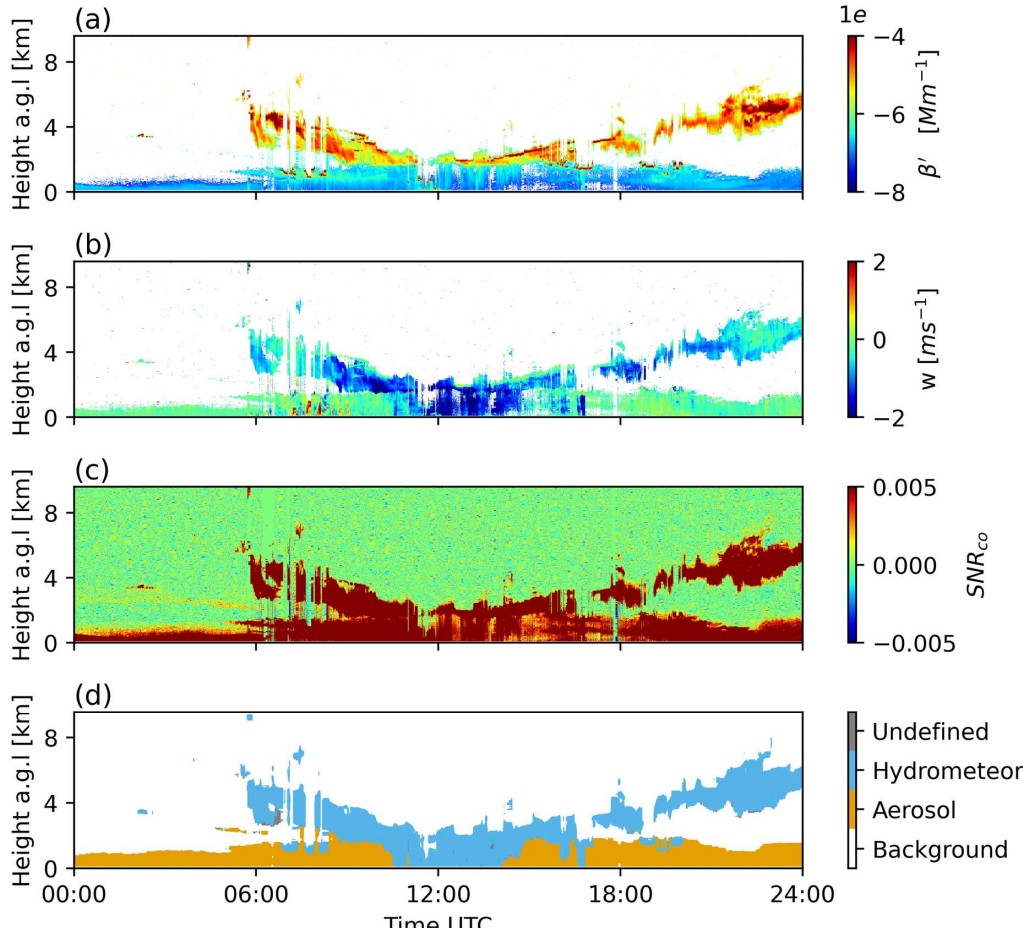

**Figure 3: Data on 12th of August 2018 at Hyytiälä. a) Attenuated backscatter (β'), b) Vertical velocity (w), c) Signal-to-noise ratio in the co-polar channel ($SNR_{co}$), d) result from AI algorithm. Data points with $SNR_{co}$ less than 3 times the background standard deviation of $SNR_{co}$ has been filtered out for [a, b, c].**

1.  The first step of the algorithm involves detecting potential hydrometeors and aerosols from background signals based on β' and $SNR_{co}$. Result from this step is shown in Figure S3a

   o  All data points with $SNR_{co}$ larger than one standard deviation of the background $SNR_{co}$ and with β' values less than $10^{-5.5}$ $Mm^{-1}$ are marked as aerosol. A 2D median kernel was then convolved with these aerosol data points to remove noisy signals due to instrumentation and attenuation.





- o  All data points with $SNR_{co}$ larger than three standard deviations of the background $SNR_{co}$ and with $\beta' > 10^{-5.5}$ $Mm^{-1}$ were marked as hydrometeor. A 2D median and maximum kernel were then convolved with these hydrometeor data points to remove noisy signals due to instrumentation and attenuation.

2. The falling hydrometeor detection step involves separating aerosol in downdrafts due to boundary layer mixing from precipitation using both $\beta'$ and $w$. Regions containing both up- and down- drafts are considered to be characteristic of boundary layer mixing, while a region of continuous downdrafts indicates precipitation. The result from this step is shown in Figure S3b

    - o  The updraft proxies are identified by selecting data points that have $w > 1$ $m\ s^{-1}$. 2D median and maximum kernels were then convolved to remove noisy signal and expand the updraft proxies.
    - o  Next, precipitation proxies are selected, having $\beta' > 10^{-7}$ $Mm^{-1}$ and $w < -1$ $m\ s^{-1}$. A 2D median kernel was then convolved with these data points to remove noisy signals. The precipitation proxies that are in the updraft proxies are then removed.
    - o  All-precipitation regions are identified having $\beta' > 10^{-7}$ $Mm^{-1}$ and $w < -0.5$ $m\ s^{-1}$. A 2D median kernel was then convolved with these data points to remove noisy signals.
    - o  The updraft region is identified having $w > 0.2$ $m\ s^{-1}$. A 2D maximum kernel was then convolved with these data points to increase the size of the updraft region.
    - o  Finally, precipitation data points are identified by including the precipitation proxies that overlap with any all-precipitation regions but not with updraft regions. A 2D maximum kernel was used for this iterative process.

3. An attenuation correction step sets all observations above clouds and precipitation with their corresponding class since the signal has been heavily attenuated. The result from this step is shown in Figure S3c

4. In the final step, a fine-tuned aerosol process is utilized to improve the aerosol class determination accuracy. The final result is shown in Figure 3d

    - o  First, aerosol clusters are identified in the time and height domain using the Density-Based Spatial Clustering of Applications with Noise (DBSCAN) algorithm (Ester et al., 1996).
    - o  A cluster is flagged as hydrometeor if the mean $w$ of the cluster $< -0.5$ $m\ s^{-1}$.
    - o  The rest of the clusters which are connected to the ground are classified as aerosol.
    - o  The rest of the clusters that have $w > -0.2$ $m\ s^{-1}$ are classified as aerosol.
    - o  All clusters that do not satisfy of the previous criteria are classified as undefined.

## 2.3 Post-processing

Aerosol is expected to be well-mixed within each aerosol layer, so in order to extract weak aerosol signal and minimize the random noise, $SNR_{co}$ and $SNR_{cross}$ were averaged for 1 hour before calculating $\delta_{aerosol}$. As mentioned before, the SNR data in





this study have been processed with the background correction algorithm described by Vakkari et al., (2019). However, at times a weak $2^{nd}$ order polynomial shape (c.f. Manninen et al., 2016) appears in $SNR_{co}$ and $SNR_{cross}$, when profiles with extended averaging (more than 1 hour) are investigated (Figure 4a). This could greatly affect the $\delta_{aerosol}$ retrieval in aerosol

layers with low SNR. Previously (Vakkari et al., 2021; Bohlmann et al., 2021), this component of the noise floor has been accounted for through a fit to SNR profiles, where hydrometeor and aerosol signals have been filtered out through visual inspection. However, given the large number of profiles analysed in this study, this approach is not feasible and thus we have automated the fitting of the $2^{nd}$ order polynomial, as described below.

For a successful fit, aerosol- and hydrometeor- free (background or noise-only) range gates need to be identified in the profile.

Firstly, the data is averaged hourly, and the profile SNR is then decomposed by stationary wavelet transform with wavelet bior2.6. Next, the variance of the noise in the SNR is removed by applying a hard threshold shrinkage function using universal thresholding (Donoho and Johnstone, 1994) to the approximation and detail coefficients from level 1 to 4. The SNR is then reconstructed using inverse stationary wavelet transform. Finally, the background range gates are identified as those having reconstructed $SNR_{co}$ values less than the standard deviation of the instrument noise floor divided by the squared root of the

number of profiles averaged in that hour. The 2nd order polynomial fit can then be performed on those noise-only data points identified in the profile (Equation 2) and the correction applied to the entire SNR profile, similar to Vakkari et al. (2019).

$$SNR_r = a + h_r \cdot b + c \cdot h_r^2 , \qquad (2)$$

where $SNR_r$ and $h_r$ are the background SNR and height at each range gate, r, and $a, b, c$ are the parameters of the fit for each profile. Overall, the effect of background correction is substantial only for low SNR values as demonstrated in section S2.

Next, the aerosol range gates are identified based on the result of the AI algorithm applied to the original non-averaged data. Finally, following Vakkari et al. (2021), the bleed-through corrected $\delta_{aerosol}$ is calculated as

$$\delta_{aerosol} = \frac{SNR_{cross} - B \cdot SNR_{co}}{SNR_{co}} , \qquad (3)$$

where $B$ is the estimated bleed through obtained from section 2.1.2.

The uncertainty of the resulting $\delta_{aerosol}$ is presented in section 2.3.1.

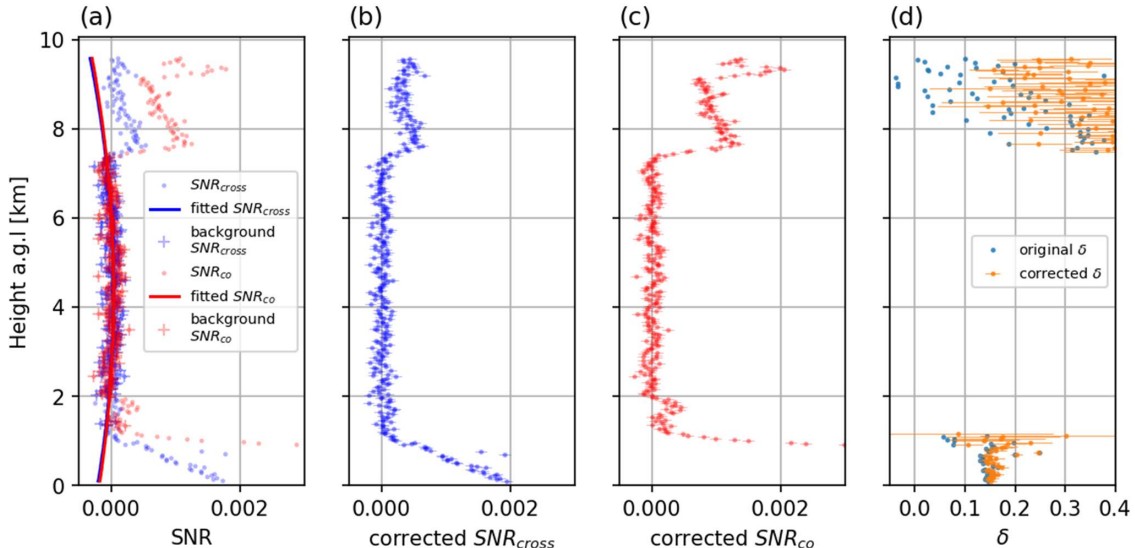

**Figure 4: Results of background correction on 2019-08-30 at 13:00 to 14:00 UTC at Hyytiälä with instrument 46. a) Profiles of co-**
**and cross-polar signal-to-noise ratio ($SNR_{co}$ and $SNR_{cross}$) , where aerosol- and hydrometeor-free regions and the corresponding 2nd**
**order polynomial fits are indicated. b) Corrected $SNR_{cross}$ profile with error bar representing the fitting uncertainty. c) Corrected**
**$SNR_{co}$ profile with error bar representing the fitting uncertainty. d) δ profile before and after the 2nd order fit correction. The**
**uncertainty in $δ_{aerosol}$ is described in section 2.3.1.**

### 2.3.1 Uncertainty in $δ_{aerosol}$

The background and the bleed through corrections include uncertainties that will propagate through to the uncertainty in the

final retrieved $δ_{aerosol}$. A Bayesian approach utilizing Stan (Carpenter et al., 2017) has been used to estimate these uncertainties.

The full Bayesian inference is obtained through a sampling method based on Hamiltonian Monte Carlo simulation (Betancourt

and Girolami, 2015), which is a form of Markov chain Monte Carlo sampling. The resulting posterior distribution is used to

estimate the mean and uncertainty of the parameters and generate the posterior predictive distribution of the variables.

The 2nd order polynomial model of SNR is similar to equation (2), but with the error component added:

$$SNR_r = a + h_r \cdot b + c \cdot h_r^2 + \epsilon , \qquad (4)$$

where $\epsilon \sim Normal(0, \sigma^2)$.

Equation (4) is equivalent to $SNR_r \sim Normal(a + h_r \cdot b + c \cdot h_r^2, \ \sigma)$, or the so-called likelihood $p(SNR \mid a, b, c , \sigma, h_r)$. The

standard non-informative prior was used for all parameters:

$$p(a) \sim Normal(0, 10^3)$$
$$p(b) \sim Normal(0, 10^3)$$
$$p(c) \sim Normal(0, 10^3)$$



$$p(\sigma)\sim Gamma^{-1}(0.001, 0.001)$$

The posterior distributions of the parameters are then obtained by sampling through Stan.

$$p(a, b, c, \sigma | SNR, h) \propto p(SNR | a, b, c, \sigma)\, p(a, b, c, \sigma)$$

Four different sampling chains are utilized with 2000 samples for each chain (excluding the 500 warmup samples), and convergence diagnostics have been carried out to ensure that the chains converged. The sampling results represent the posterior distribution of all the parameters.

From these posterior distributions, the mean and standard deviation of the parameters can be calculated. The fitted background
SNR distribution at all heights can be generated by sampling from the posterior predictive distribution. It is done through selecting random draws from the posterior, which are then plugged into equation (4) to calculate the posterior predictive samples. The mean and standard deviation of the fitted background SNR at each height can be estimated by calculating the mean and standard deviation of the generated samples at each height.

The samples of the corrected SNR distribution at each height are calculated by dividing the original SNR by the samples from
the estimated background SNR distribution at each height. The samples of the final $\delta_{aerosol}$ distribution at each height are then calculated following equation 2 assuming the bleed-through follows a normal distribution with mean and standard deviation estimated from section 3.1.2. The mean and uncertainty of the final $\delta_{aerosol}$ is estimated by calculating the mean and standard deviation of the samples at each height.

## 2.4 Air mass origin

For two case studies, the air mass origin was investigated using the Lagrangian transport model Flexpart version 10.4 run in backwards mode (Seibert and Frank, 2004; Stohl et al., 2005; Pisso et al., 2019). Flexpart was run using ERA5 reanalysis obtained from the European Centre for Medium-Range Weather Forecasts (ECMWF) as the meteorological input. ERA5 was obtained with a temporal resolution of 1 hour and a spatial resolution of 0.25° for a domain covering 125° W to 75° E and 10° N to 85° N. Vertical model levels 50-137 were retrieved, which includes approximately the lowest 20 km a.g.l. For the elevated
layers considered in the case studies, Flexpart was run backwards in time for 7 days and potential emission sensitivity (PES) was saved with 1-hour temporal, 0.2° latitude-longitude resolution. The output resolution in the vertical was 250 m from ground level to 5 km a.g.l. Above 5 km a.g.l, output levels at 10 km a.g.l and 50 km a.g.l were included. PES is proportional to the air mass residence time in a grid cell and was obtained in units of seconds (Seibert and Frank, 2004; Pisso et al., 2019).





# 3 Results and discussion

## 3.1 Instrumental performance

### 3.1.1 Noise floor level

Figure 5a displays the noise floor level for each instrument, calculated as the standard deviation of the $SNR_{co}$ ($\sigma_{SNR}$) in the aerosol- and hydrometeor- free region. As expected, integration time is one important factor with, for example, instrument Utö-32 showing a dramatic increase in the noise floor level in summer 2016 when the integration time was decreased (from 27.5s to 5s). Since each instrument may have its own configuration, normalization of the noise floor level considering such differences is required in order to compare the noise floor between instruments.

Figure 5b illustrates the noise floor levels calculated after scaling for number of pulses in each integration time and demonstrates that most instruments remained stable during the study period. Fluctuations were observed in the noise floor level for the old instrument Utö-32, but the noise floor level itself is deemed to be low, with visual inspection indicating no issues in the data quality. Sodankylä-54 is the only instrument that shows a systematic increase in the noise floor level over time. This is a worrying sign, but so far, the increase in the noise floor has been relatively small at approximately 2% from 2017 to 2019. However, further increases in the noise floor will begin to limit the retrieval of weak signals.

The StreamLine Pro lidar instruments, Vehmasmäki-53 and Sodankylä-54 have similar and systematically higher noise floor levels than the other instruments. This is due to the Streamline Pro models were configured to utilize only half of the bandwidth, i.e half the Nyquist velocity. The Utö-32XR instrument, which is the Utö-32 system after being upgraded with a StreamLine XR transmitter and receiver, has the lowest noise floor of all instruments in this study.

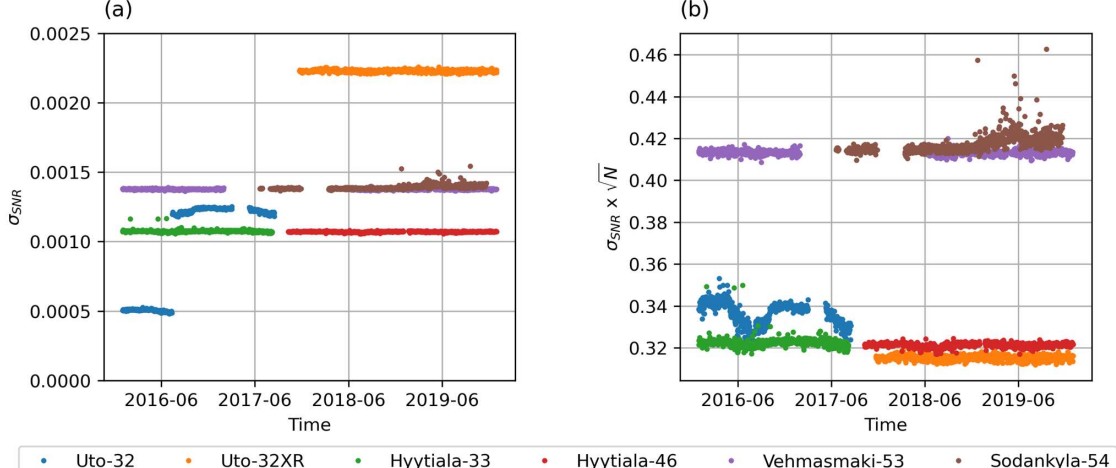

**Figure 5: (a) Standard deviation of the background co-polar signal-to-noise ratio ($\sigma_{SNR}$). (b) The same but scaled with number of**
**pulses in each integration (denoted with N).**

### 3.1.2 Bleed through

Figure 6 displays the time series of $\delta$ at liquid cloud base for each instrument in the network. For Utö-32 (Figure 6a), the
overall mean $\delta$ is the highest of the instruments in this study. This probably stems from its long integration time configuration
(see Table 2), resulting in $SNR_{co}$ and $SNR_{cross}$ not measuring the same part of the cloud.

In addition, the occurrence of liquid cloud varies in time and with location. Mixed-phased clouds are difficult to distinguish
from liquid-only clouds, hence, the high value of $\delta$ at liquid cloud base especially during wintertime (Figure 6) might be due
to the misclassification with mixed-phased clouds. This effect is more prominent in Utö-32 and Hyytiälä, leading to longer
tailed distributions (Figure 6c and 6d) than at other sites. On the other hand, for Utö-32XR, Vehmasmäki-53 and Sodankylä-
54 (Figure 6b, 6e and 6f), the $\delta$ at the collected liquid cloud base varies much less and remain stable at a low level ($\delta$ <0.01)

throughout the year.

To quantify the bleed through of the internal polarizer, the elevated value of $\delta$ due to imperfect sampling of pure liquid clouds
need to be excluded. Here, we assume that for Utö-32, Hyytiälä-33 and Hyytiälä-46, the tail of the distribution due to mixed-
phase clouds or non-ideal sampling of liquid clouds, whilst the peak of the distribution is from the liquid cloud base. A Gaussian
mixture model, which is a variant of finite mixture models (Baek et al., 2010), was used to derive the mean and standard

deviation of $\delta$ at the liquid cloud base ignoring the tail of the distribution. For most instruments, excluding the tail of the
distribution in Figure 6 has a minimal effect on the average bleed-through estimate, which remains around 1 % as shown in
Table 3. The largest effect is seen for Hyytiälä-46, where including all $\delta$ values at liquid cloud base results in a bleed-through
estimate $0.020 \pm 0.022$, while excluding the tail decreases the bleed-through estimate to $0.008 \pm 0.007$. Overall, the values of





the best-estimate bleed-through in this study are smaller than the previous short case study estimates of 0.011 ± 0.007 in
Limassol (Vakkari et al., 2021); 0.016 ± 0.009 and 0.013 ± 0.006 in Vehmasmäki (Vakkari et al., 2021; Bohlmann et al., 2021)
for similar instruments.

**Figure 6: Time series (left panels) and histograms (right panels) of depolarization ratio (δ) at liquid cloud base in a) Utö-32, b) Utö-32XR, c) Hyytiälä-33, d) Hyytiälä-46, e) Vehmasmäki-53, f) Sodankylä-54. The best estimates of mean (μ) and standard deviation (μ) of δ at the liquid cloud base in each site are shown in the right panel.**



| Instrument | Best estimate of bleed-through | Bleed-through including all data |
|---|---|---|
| **Utö-32** | $0.011 \pm 0.007$ | $0.018 \pm 0.018$ |
| **Utö-32XR** | $0.004 \pm 0.004$ | $0.004 \pm 0.004$ |
| **Hyytiälä-33** | $0.005 \pm 0.005$ | $0.009 \pm 0.011$ |
| **Hyytiälä-46** | $0.008 \pm 0.007$ | $0.020 \pm 0.022$ |
| **Vehmasmäki-53** | $0.007 \pm 0.007$ | $0.007 \pm 0.007$ |
| **Sodankylä-54** | $0.005 \pm 0.005$ | $0.004 \pm 0.005$ |

**Table 3: Bleed-through (mean and standard deviation of depolarization ratio at the liquid cloud base) for each instrument**

### 3.2 Comparing Aerosol Identification algorithm with Cloudnet classification

Cloudnet's goal is to provide continuous ground-based observations of the cloud properties for forecast and climate model (Illingworth et al., 2007). On the other hand, the primary aim of the Aerosol Identification (AI) algorithm developed here is to exclude hydrometeors from the aerosol mask to ensure that aerosol characteristics are not biased by signal from hydrometeors.

Of all the sites in this study, only Hyytiälä belongs to the Cloudnet's network. There, the result of the AI algorithm based on Doppler lidar only (AI algorithm, Sect. 2.2) can be compared with the classification from Cloudnet (Illingworth et al., 2007; Tukiainen et al., 2020). However, there are marked differences between the two algorithms. The Cloudnet classification algorithm utilises a combination of several instrument, most importantly being lidar and cloud radar (Illingworth et al., 2007). The backscattered signal from radars is much more sensitive to the diameter (D) of the particle, $\sim D^6$ (Rauber and Nesbitt, 2016) compared to lidar with only $\sim D^2$ (Weitkamp, 2005). As the result, radar's signal is dominated by larger particles while lidar's is dominated by smaller particles with higher concentration. Additionally, the extinction of radar signal in non-precipitating clouds is negligible, so it can observe much further into the clouds.

Figure 7 presents a side-by-side comparison example of the results from these algorithms. At first glance, it is obvious that Cloudnet underestimates the extent of the aerosol layer compared to the AI algorithm. This is due to the improved post-processing applied to the Doppler lidar data, which allows the AI algorithm to capture data from weaker aerosol signal than Cloudnet. Unsurprisingly, cloud mask in Cloudnet classification includes the full cloud layer, while AI algorithm indicates often only the cloud base due to the attenuation of lidar signal. There are some minor differences in the precipitation zone such as right before 18:00 UTC in Figure 7a and 7b and at 18:00 UTC in Figure 7e and 7f, when some parts of the precipitation in Cloudnet are flagged as aerosol in the AI algorithm. This is probably due to the differences in sensitivity of different particle size in cloud radar and lidar as discussed previously. Figure 7g and 7h display a snow event from 0:00 UTC to around 11:00

UTC. Cloudnet shows that some parts of snowfall eventually melt into raindrops near the ground, whilst the AI algorithm also correctly classified the whole event as hydrometeor.

After comparing all the data in Hyytiälä throughout the whole study period, we found that 7.7% of the aerosol data points from the AI algorithm is classified as hydrometeor in Cloudnet algorithm. This result shows that the AI algorithm performs

adequately in extracting aerosol data comparing to Cloudnet classification algorithm. The differences between these two algorithms stem from differences in goals and instrumentations. Given the large amount of data, we are confident that the AI algorithm is capable of extracting the overall statistics of $\delta_{aerosol}$ for the purpose of this study.

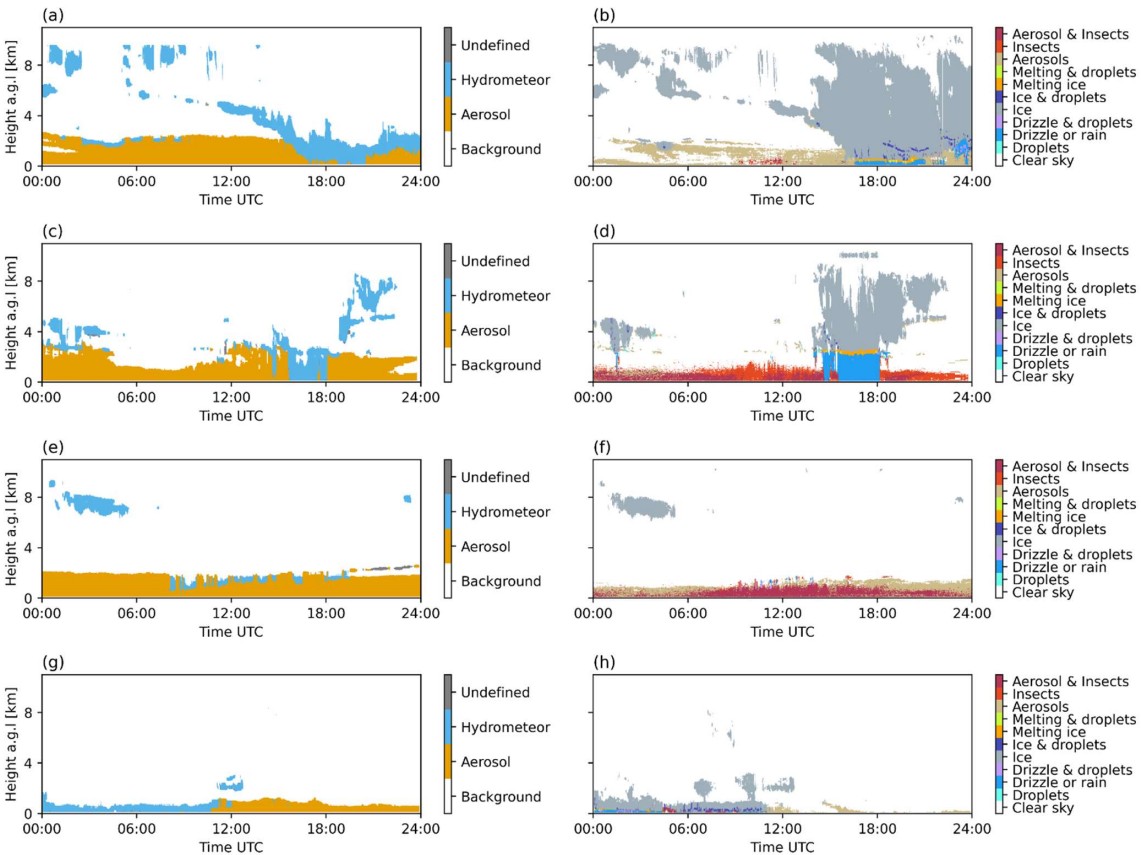

**Figure 7: Results in Hyytiälä from the aerosol identification algorithm and Cloudnet classification algorithm respectively on [a, b] 2018-06-11; [c, d] 2019-04-07; [e, f] 2019-08-07; and [g, h] 2019-12-31.**





**3.4 Aerosol particle depolarization ratio**

Here, we analyse $\delta_{aerosol}$ from the four-year observational data set at four sites in Finland. The $\delta_{aerosol}$ was obtained following the post-processing procedure described in Section 2.3 and the AI algorithm was used to exclude hydrometeors from the data

set. All data that have standard deviation of $\delta_{aerosol}$ larger than 0.05 have been filtered out.

| Sites | 25th percentile | 50th percentile | 75th percentile | Mean | Standard deviation |
|---|---|---|---|---|---|
| Utö | 0.03 | 0.07 | 0.12 | 0.09 | 0.07 |
| Hyytiälä | 0.06 | 0.11 | 0.18 | 0.13 | 0.08 |
| Vehmasmäki | 0.05 | 0.10 | 0.16 | 0.11 | 0.07 |
| Sodankylä | 0.01 | 0.05 | 0.11 | 0.07 | 0.08 |

**Table 4: Overall statistics of $\delta_{aerosol}$ across all the sites**

Overall, as seen in Table 5, the average $\delta_{aerosol}$ are higher in the boreal forest sites of Hyytiälä (0.13 ± 0.08) and Vehmasmäki (0.11 ± 0.07) compared to Utö (0.09 ± 0.07) and Sodankylä (0.07 ± 0.08). This can be explained by the abundance of pollen

with high $\delta_{aerosol}$ in the boreal forest sites (Aaltonen et al., 2012; Vakkari et al., 2021; Bohlmann et al., 2021; Shang et al., 2020). Located in the marine site, the lidar at Utö observes a high fraction of marine aerosols with low $\delta_{aerosol}$ as also found in previous studies (Vakkari et al., 2021; Haarig et al., 2017a; Mylonaki et al., 2021; Groß et al., 2013, 2011) . In Sodankylä, which is a clean rural subarctic environment, $\delta_{aerosol}$ is found to be the lowest of the sites considered here.

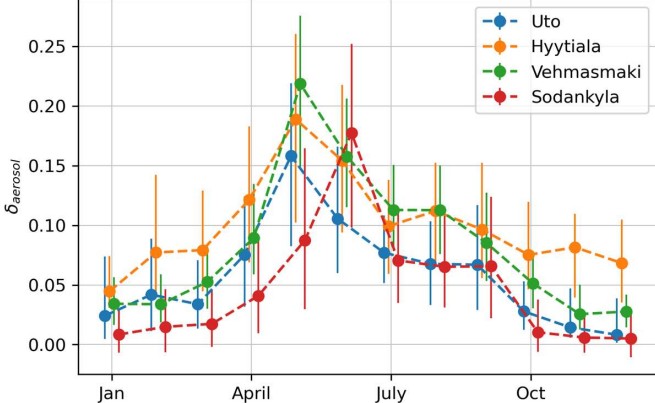

**Figure 8: Monthly median of aerosol depolarization ratio ($\delta_{aerosol}$) across all the sites, error bars show the 25th and 75th quantile**

Figure 8 shows the average monthly median of $\delta_{aerosol}$ for all sites. Overall, $\delta_{aerosol}$ is increased in the summer and remains low in the winter. The enhanced values of $\delta_{aerosol}$ from April to the end of September coincide with the growing season of the boreal



forest (Manninen et al., 2014) when airborne pollen and other particles of biological origin are abundant in Finland. The most common sources of pollen in the Finnish boreal forest are birch (Betula), spruce (Picea), pine (Pinus), and nettle (Urtica). They

contribute to more than 90% of the total pollen load (Shang et al., 2020; Manninen et al., 2014). The peaks of $\delta_{aerosol}$ occurs in the pollen season of birch, spruce, and pine (Shang et al., 2020) between May and early June. The irregular shape of these pollen types has been observed to have high $\delta$ such as $0.10 \pm 0.06$ for birch at 532nm (Bohlmann et al., 2019), $0.29 \pm 0.10$ for spruce at 1565nm (Bohlmann et al., 2021), and $0.36 \pm 0.01$ for pine at 532nm (Shang et al., 2020). The relatively high concentration of these pollen grains in the atmosphere likely explains the peak of $\delta_{aerosol}$ in May for the boreal forest sites of

Hyytiälä and Vehmasmäki.

However, the peak in $\delta_{aerosol}$ at Sodankylä occurs one month later than the other sites. Due to its location in the sub-arctic, the snow melt and the beginning of the pollen season start later at around June in Sodankylä than at other sites in central and southern Finland (Oikonen et al., 2005; Koivikko et al., 1986). Interestingly, despite its marine location, $\delta_{aerosol}$ in Utö also peaks in May and the $\delta_{aerosol}$ value remains high during summer, probably due to transported pollen from the mainland. Even

though pollen grains are typically large from 10 to 100µm (Manninen et al., 2014), they are low-density particles, which make them prone to be lifted by turbulent air flows up to several kilometres and be dispersed by the wind (Bohlmann et al., 2019). Several studies (Szczepanek et al., 2017; Skjøth et al., 2007; Rousseau et al., 2003, 2006) have demonstrated the long-range transport of pollen over thousands of kilometres. Additionally, marine aerosol at lower relative humidity in the summer (see Figure 10) could contribute to the higher $\delta_{aerosol}$ in the summer at Utö. Haarig et al., (2017a) found that at low relative humidity,

elevated marine aerosol can crystalize and become mostly cubic-like in shape (Wise et al., 2007), which results in a higher $\delta_{aerosol}$ up to $0.15 \pm 0.03$ at 532nm and $0.1 \pm 0.01$ at 1064nm.

During the winter months from October to March, $\delta_{aerosol}$ remains low across all sites. Luoma et al., (2019) found that there is a high fraction of aerosol from anthropogenic sources and domestic wood burning in the winter in Hyytiälä, and these are probably major sources of aerosol in other continental sites as well. Burton et al., (2015) reported the $\delta_{aerosol}$ of smoke as 0.019

$\pm 0.005$ at 1064nm, and with its negative wavelength dependency (Ohneiser et al., 2020; Haarig et al., 2018), $\delta_{aerosol}$ of smoke at 1565nm would probably have even smaller values. Anthropogenic aerosol has also been found to have small $\delta_{aerosol}$ values < 0.05 at 532nm (Burton et al., 2013; Müller et al., 2007; Mylonaki et al., 2021). The low $\delta_{aerosol}$ in Utö < 0.05 agrees with the typical $\delta_{aerosol}$ of wet and polluted marine aerosol found at various wavelengths (Groß et al., 2011; Vakkari et al., 2021; Haarig et al., 2017a; Mylonaki et al., 2021).






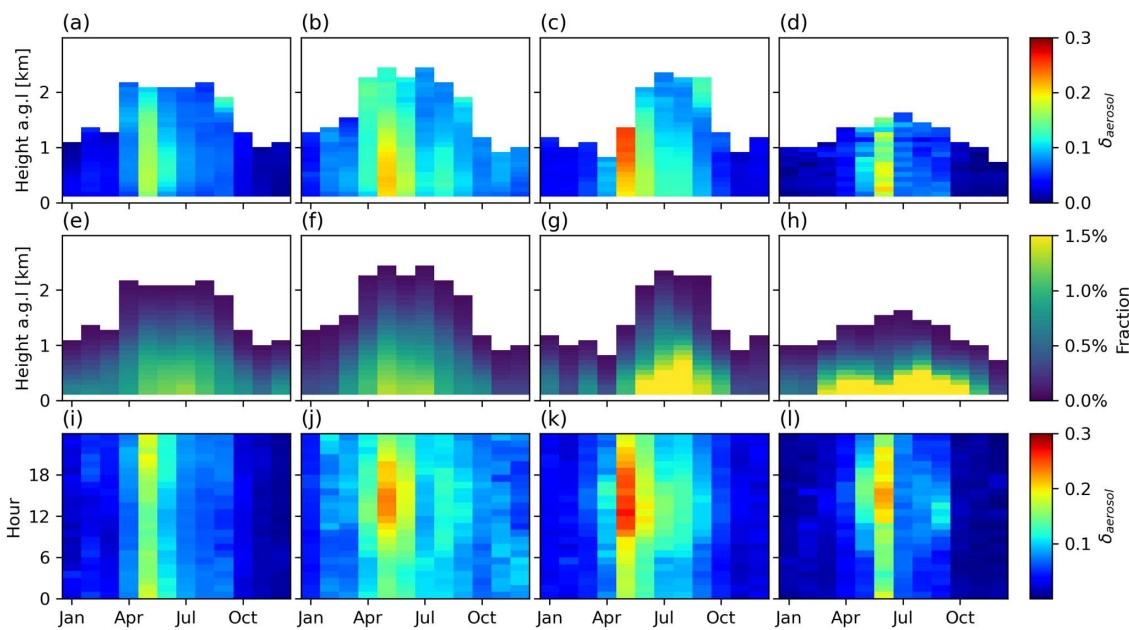

**Figure 9: Median of aerosol depolarization ratio (δ_aerosol) profiles across all the sites. Utö [a, e, i]; Hyytiälä [b, f, j]; Vehmasmäki [c, g, k]; and Sodankylä [d, h, l]. Distribution of δ_aerosol with height ratio in each month is shown on the first row [a, b, c, d]. The frequency of observations relative to height in each month is shown on the second row [e, f, g, h]. The last row [i, j, k, l] displays the diurnal pattern of a δ_aerosol in each month.**


Figure 9a, 9b, 9c and 9d display the average $\delta_{aerosol}$ as a function of height for each site. During summer from April to August, aerosols with high $\delta_{aerosol}$ are concentrated within the boundary layer, often below 1 km a.g.l. However, in Vehmasmäki during May, $\delta_{aerosol}$ increases with height, the $\delta_{aerosol}$ value can reach up to 0.3 at 1.2km a.g.l. As mentioned, pollen grains can get easily lofted to higher altitude and since the air is dryer at high altitude, pollen might fold onto itself to prevent further

dehydration which would lead to higher $\delta_{aerosol}$ (Bohlmann et al., 2019).

In September at Utö, Hyytiälä and Vehmasmäki, aerosol with $\delta_{aerosol} > 0.1$ can be observed at around 2 km a.g.l. This is probably due to transported aerosol since the $\delta_{aerosol}$ is much higher than boundary layer aerosol near the ground. During winter months from October to March, $\delta_{aerosol}$ values were distributed uniformly at all heights for every site.

Figure 9e, 9f, 9g and 9h show the frequency of aerosol detected at each height level across all sites. Overall, more aerosol is

observed in the summer months, and at higher altitude, due to the higher boundary layer height in summer. In Sodankylä, aerosol is observed less frequently at altitudes above 1 km a.g.l compared to other sites.

Figure 9i, 9j, 9k and 9l illustrate the diurnal pattern of $\delta_{aerosol}$ across all sites. During May-June, when monthly $\delta_{aerosol}$ is at its highest, median $\delta_{aerosol}$ in Hyytiälä, Vehmasmäki and Sodankylä peaks in the afternoon. The release of pollen is positively





influenced by higher temperature (Bartková-Ščevková, 2003), so the higher $\delta_{aerosol}$ during daytime might be due to higher

fraction of irregular shaped pollen being released at noon. Simultaneously, higher mixing layer height during afternoon hours enables more pollen grains released at the surface to be lifted above the lidar's minimum range. This result also agrees with the previous findings (Noh et al., 2013a; Bohlmann et al., 2021), where the high $\delta_{aerosol}$ from pollen is only observed in the boundary layer during daytime.

On the contrary, during May in Utö $\delta_{aerosol}$ peaks at night and remains low during the day (Figure 9i). We attribute this high

value of $\delta_{aerosol}$ to transported pollen arriving from the continent at night, which may be part of the returning flow of a sea breeze circulation in the Baltic Sea (Dailidė et al., 2022).

### 3.4.1 The effect of relative humidity

Studies show that relative humidity (RH) can change the shape of marine aerosol (Granados-Muñoz et al., 2015; Haarig et al.,

2017a) and pollen grains (Franchi et al., 1984; Griffiths et al., 2012; Katifori et al., 2010) through hygroscopic growth or rupture into smaller fragments (Hughes et al., 2020; Miguel et al., 2006; Taylor et al., 2002, 2004; Bohlmann et al., 2021). Hence, $\delta_{aerosol}$ at the sites in this study may also respond to the ambient RH. However, as RH profiles are not measured, we investigate the connection between surface RH and $\delta_{aerosol}$ below 300 m a.g.l..

Figure 10 shows the seasonal and diurnal pattern of $\delta_{aerosol}$ and RH below 300 m a.g.l. The $\delta_{aerosol}$ pattern of the closest 300m

(Figure 10e, 10f, 10g and 10h) is similar to that of the whole profile (Figure 9i, 9j, 9k, 9l) in that its value is highest in the summer and lowest in winter. During May and June, the continental sites present a strong diurnal variation in both RH and $\delta_{aerosol}$. However, the island of Utö also shows a similar if less pronounced diurnal cycle in RH to the continental sites, but an opposite diurnal cycle response for $\delta_{aerosol}$ as noted before due to transported pollen.

Previously, pollen concentrations have been shown to be enhanced around noon when RH is at its lowest (Käpylä, 1984;

Latorre and Caccavari, 2009, Noh et al., 2013b). In fact, pollen release is highly dependent on the particular ambient weather conditions including rainfall, air temperature, RH, duration of sunshine and wind speed (Adams-Groom et al., 2002; Alba et al., 2000; Bartková-Ščevková, 2003; Gilissen, 1977; Vázquez et al., 2003; Jato et al., 2000; Käpylä, 1984; Noh et al., 2013b). Moreover, low RH could affect the diffusion of pollen in the atmosphere by decreasing the specific gravity of pollen grains, thus negatively reduce their ability to settle to the ground (Durham, 1943), i.e., be more easily diffused above the lidar

minimum range of 90 m a.g.l.





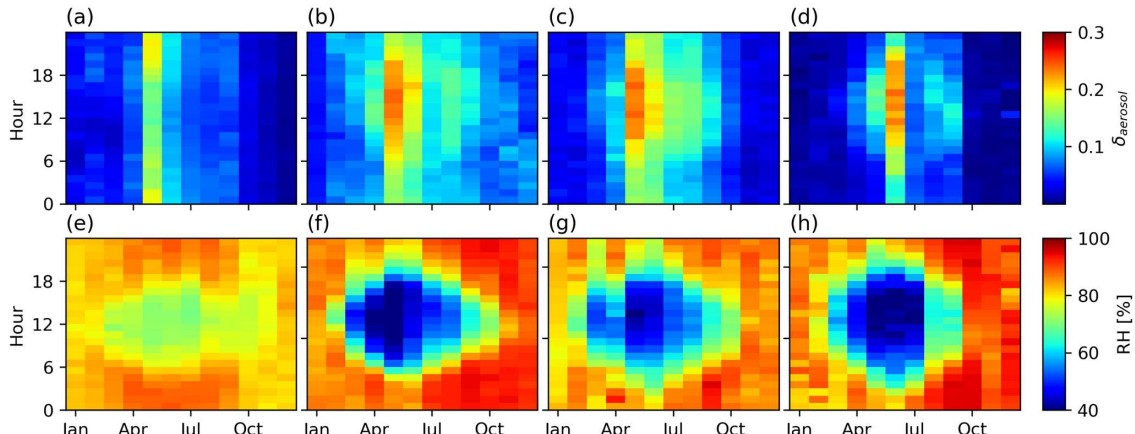

**Figure 10: Diurnal pattern of aerosol depolarization ratio (δ_aerosol) and relative humidity (RH) below 300m a.g.l respectively in Utö [a, e], Hyytiälä [b, f], Vehmasmäki [c, g], Sodankylä [d, h]**

Statistical analysis (Table S4) shows an overall negative correlation between RH and $\delta_{aerosol}$. However, without additional data,

attributing the change of aerosol physical properties such as hygroscopic growth as the reason for this negative correlation is not possible. The change of aerosol type such as from pollen in the summer to anthropogenic aerosol in the winter, or diurnal changes in pollen release could be the main driver of $\delta_{aerosol}$.

### 3.5 Case studies

### 3.5.1 Hyytiälä April 2018

From late night 14 April 2018 to morning 15 April 2018, an elevated aerosol layer between 2 km to 4 km a.g.l. was observed above Hyytiälä (Figure 11). The averaged profile from 03:00 to 04:00 UTC on 15 April 2018 shows an elevated aerosol layer with $\delta_{aerosol} = 0.24 \pm 0.008$, while in the boundary layer (from the surface to 1.5 km a.g.l.), $\delta_{aerosol} = 0.12 \pm 0.004$. This substantial difference in $\delta_{aerosol}$ indicates that the elevated aerosol had likely undergone long-range transport to the site. The FLEXPART simulation of this layer (Figure S5) indicates that the air mass was not in contact with the surface for 4.5 days prior to arrival

over Hyytiälä. However, 5-7 days before arrival, a contribution from the surface layer was seen coming mostly from the western Sahara, with minor contributions from the Mediterranean (Figure S5). This, together with the high $\delta_{aerosol}$, suggests the layer to be mostly Saharan dust; the $\delta_{aerosol}$ value is comparable to values reported in the literature at other wavelengths (Groß et al., 2011; Freudenthaler et al., 2009; Burton et al., 2015; Haarig et al., 2017b). However, it is slightly lower than the dust value of 0.30 reported by Vakkari et al., (2021) at 1565nm. This could be due to the decrease of $\delta_{aerosol}$ in long range transport

dust layer, resulted from gravitational settling of large dust particles during long range transport (Haarig et al., 2017b).



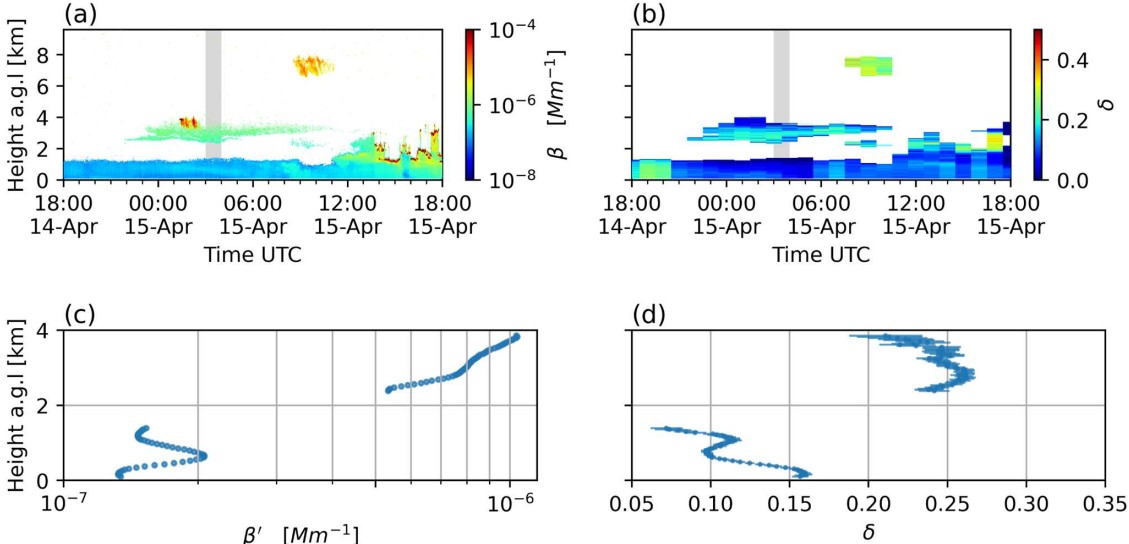

**Figure 11: Profiles on 2018-04-14 to 2018-04-15 at Hyytiälä a) Attenuated backscatter (β'), b) 1-hour averaged depolarization ratio (δ). And 300m running mean (in shaded area) on 2018-04-15 03:00-04:00 UTC of c) β' profiles c) δ$_{aerosol}$ profiles**

### 3.5.2 Utö May 2017

On 13 May 2017, an elevated aerosol layer between 800 m a.g.l and 2 km a.g.l was observed over Utö (Figure 12). For the period on 2017-05-13 from 18:00 to 20:00 UTC, δ$_{aerosol}$ of this elevated layer is at $0.23 \pm 0.01$ and decreases with height (Figure 12d and 12f). At the same time, δ$_{aerosol}$ of the boundary layer aerosol from the surface up to 500 m is at $0.26 \pm 0.009$.

The air mass history shows that this layer originated from both offshore in the Baltic Sea and from continental Finland (Figure S6). The layer was elevated from the surface only 1 day before arrival and remained at a height of 1-2 km a.g.l. In this region

there are no other known sources of aerosol with such high δ$_{aerosol}$ except for pollen, and thus this layer is probably pollen transported from the surrounding continental areas. From 2017-05-14 00:00-05:00 UTC, this layer merged into the boundary layer, with the maximum value of δ$_{aerosol}$ in the boundary increasing to $0.30 \pm 0.007$. The value of δ$_{aerosol}$ is comparable to previous observations for spruce pollen in boreal forest; $0.269 \pm 0.005$ (Vakkari et al., 2021) and $0.29 \pm 0.10$ (Bohlmann et al., 2021).




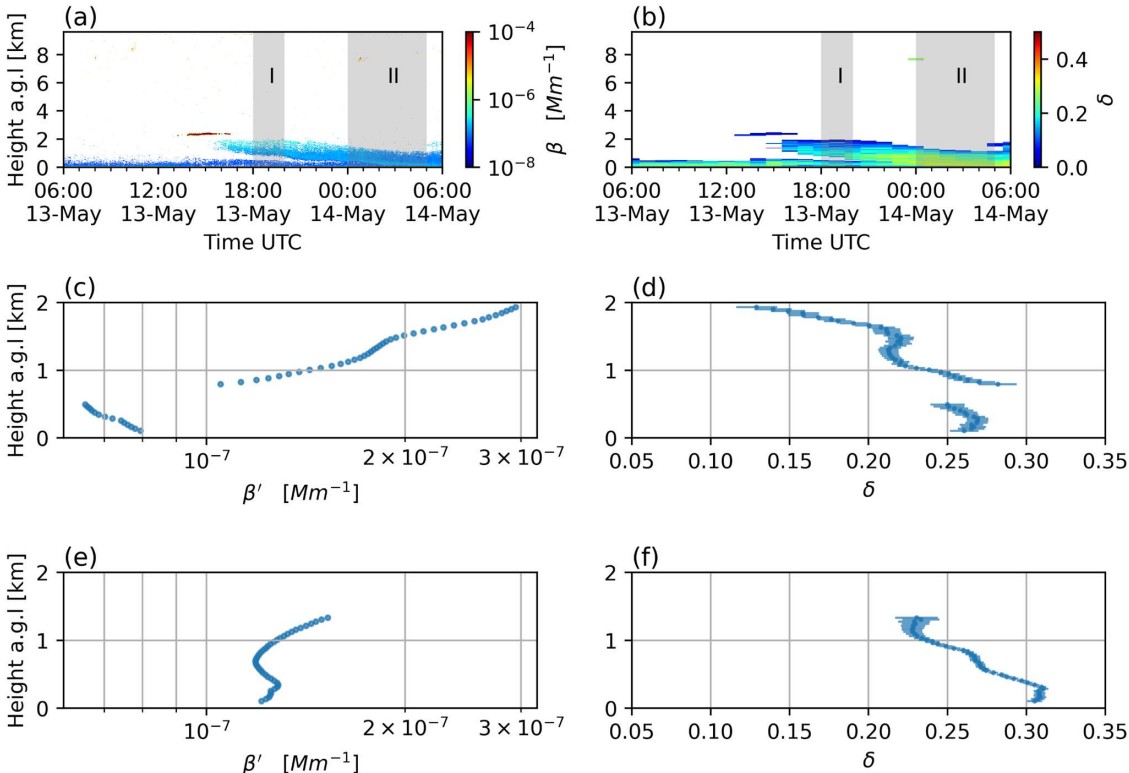

**Figure 12: Profiles on 2017-05-13 to 2017-05-14 at Utö a) Attenuated backscatter (β') b) 1-hour averaged depolarization ratio (δ). And 300m running mean (shaded area I) at 2017-05-13 18:00-20:00 UTC of c) β' profile c) δ_aerosol profile. 300m running mean (shaded area II) at 2017-05-14 00:00-05:00 UTC of c) β' profile c) δ_aerosol profile**


## 4 Conclusions

In this study, the use of Halo Doppler lidars for long-term monitoring of $\delta_{aerosol}$ was investigated with a four-year long data set from the Finnish remote sensing network. The first aim was to investigate the stability of the instrumental noise floor and internal polarizer performance. The second aim was to characterise the seasonal and diurnal variation of $\delta_{aerosol}$ at a wavelength of 1565 nm at four different sites in Finland.


The instrumental noise floor was assessed through the time series of the standard deviation of $SNR_{co}$ in noise-only (aerosol- and hydrometeor- free) conditions. Four of the six systems included in this study presented a very stable noise floor, but two systems had up to 6% variability in the $\sigma_{SNR}$ time series, even after changes in the configured integration time were taken into



account. However, visual inspection did not indicate major issues in the data from the systems with increased variability in

$\sigma_{SNR}$.

The time series of δ at liquid cloud base did not indicate long-term changes in the bleed-through of the Halo Doppler lidar internal polarizer. Overall, the observed values were comparable to previous case studies, though some elevated (δ > 0.02) values were observed especially during the wintertime, possibly due to the inclusion of some mixed-phase clouds erroneously classified as liquid-only. Also, the configuration with very long integration time of 27.5 s resulted more frequently in elevated

δ at liquid cloud base, leading to a long tail in the otherwise Gaussian distribution of δ. Excluding the tail of the distribution resulted in bleed-through estimates that are very similar for all five instruments that were included in the study, ranging from 0.004 to 0.011. Including all the δ values would increase the bleed-through estimate at most from 0.008 ± 0.007 to 0.020 ± 0.022 for the Hyytiälä-46 lidar, for other systems the effect is smaller.

From the four-year observations of $\delta_{aerosol}$, it is found that $\delta_{aerosol}$ is surprisingly similar at all four sites in Finland. The overall

$\delta_{aerosol}$ are 0.13 ± 0.08 in Hyytiälä, 0.11 ± 0.07 in Vehmasmäki, 0.09 ± 0.07 in Utö and 0.07 ± 0.08 in Sodankylä. All these sites have low $\delta_{aerosol}$ in the winter months and high $\delta_{aerosol}$ in the summer months, which is attributed to the presence of irregular-shaped pollen particles. The highest monthly averages in $\delta_{aerosol}$ are observed in May and June. During this period, $\delta_{aerosol}$ has a strong diurnal cycle: at Hyytiälä, Vehmasmäki and Sodankylä, $\delta_{aerosol}$ peaks in the afternoon, while at Utö the peak is several hours later in the night. This difference for the island station of Utö is attributed to the time lag required for pollen to be

transported from the mainland to the island. Additionally, $\delta_{aerosol}$ in the nearest 300 m a.g.l. was found to have a negative correlation with relative humidity at the surface. However, this negative correlation could stem from either hygroscopic growth or from concurrent changes in the aerosol type e.g., due to pollen release diurnal cycle. Attributing which factor plays a more dominant role is challenging and would require more profiling measurements.

Two cases of elevated, long-range transported aerosol were studied in more detail. From late night 14 April 2018 to the morning

15 April 2018, an elevated aerosol layer between 2 km to 4 km a.g.l. was observed at Hyytiälä. The unusually high $\delta_{aerosol}$ of 0.24 ± 0.008 for April at Hyytiälä for this event, together with air mass history analysis from FLEXPART indicating a source contribution from North-West Africa, mean that this layer could be identified as Saharan dust. The $\delta_{aerosol}$ value of this layer is comparable to previous observations of long-range transported Saharan dust (Groß et al., 2011; Freudenthaler et al., 2009; Burton et al., 2015; Haarig et al., 2017b; Vakkari et al., 2021). On 13 May 2017 in Utö, an elevated aerosol layer with elevated

$\delta_{aerosol}$ was observed between 800m and 2km a.g.l.. The air mass footprint of this layer was from the Baltic Sea and surrounding areas, suggesting the layer contained pollen originating from continental areas and arriving at Utö at night. During this episode the $\delta_{aerosol}$ ranged from 0.23 to 0.30, which was well within the range of previous reports for pollen in Finland (Bohlmann et al., 2019, 2021; Shang et al., 2020; Vakkari et al., 2021).

In conclusion, the long-term performance of Halo Doppler lidars was found to be satisfactory for the determination of $\delta_{aerosol}$

for continuous monitoring purposes. Furthermore, the aerosol identification algorithm developed for this study, which is based on Doppler lidar only, was found to agree with the aerosol mask from the multi-instrument Cloudnet classification algorithm


for more than 90% of time. This extends the capabilities of the Finnish remote sensing network (Hirsikko et al., 2014) in identifying potentially hazardous aerosol particles, which was one of the motivations in choosing Halo Doppler lidars for the network.


*Code availability*

The aerosol identification algorithm is available at https://github.com/vietle94/HaloDopplerLidar-AerosolIdentification.

*Data availability*

Lidar data are available upon request from the authors.

*Author contributions*

Analysis was carried out by VL and HL. Data was curated by EJOC and VV. Conceptualization and funding acquisition was carried out by VV. VL wrote the original draft, which was reviewed and edited by VV, EJOC and HL.


*Competing interests*

The authors declare that they have no conflict of interest.

*Acknowledgements*

Financial support by National Emergency Supply Agency of Finland, by Academy of Finland (grant no 337552, 343359, 346643) and by Magnus Ehrnrooth Foundation is gratefully acknowledged.

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
