# Peer review of "Long-term aerosol particle depolarization ratio measurements with Halo Doppler lidar"

_Atmospheric Measurement Techniques, 2023_

## Author Comment (AC1)

**Response to reviewers comments for AMT-2023-37**

Viet Le[1], Hannah Lobo[1], Ewan J. O'Connor[1], and Ville Vakkari[1, 2]

[1]Finnish Meteorological Institute, Helsinki, 00101, Finland
[2]Atmospheric Chemistry Research Group, Chemical Resource Beneficiation, North-West University, Potchefstroom, 2520, South Africa

**Correspondence:** viet.le@fmi.fi

We would like to thank the reviewers for the constructive comments. Below are the reviewer comments in black text, followed by our responses in blue text. Where applicable, we also provide changes to the manuscript in *blue italics*.

**1    Reviewer 1**

The manuscript submitted by Viet Le et al. presents long-term measurements with several Doppler lidars in Finland for the purpose of depolarization measurements of atmospheric aerosol.

It is one of the first studies (if not the first) presenting a long-term study of depolarization ratio of aerosol observed by a Doppler lidar – which is very valuable. The remote locations in Finland make the result rather representative for less populated areas (even if station is characterized as semi-urban) and thus I see a high potential for follow-on studies in other regions. With the techniques and methods described in the manuscript, also other HALO Doppler Lidar in different region of the world can hopefully be utilized for future studies and thus collect a valuable data set. Thus, I consider this study pioneering and paving the road for other groups to do so similar.

The paper is relevant for the scientific community and well written and mostly well organized. Argumentation is logical and reasonable and it is worth publishing. The content would have been sufficient to for 3 publications, and sometimes is very technical, but I consider all these information as very valuable.

However, I have some minor comments, which are needed to make the excellent results better understandable (and reproducible) for the broader community of AMT and not only (Doppler) lidar experts. Sometimes, after addressing these points, the manuscript can be published.

We would like to thank the reviewer for their positive evaluation of the manuscript.

General comments:

The authors should properly introduce the variables they use, i.e., the attenuated backscatter coefficient and the volume (particle) depolarization ratio. And then explain, why in case of the HALO lidar, the molecular contribution can be neglected (i.e., why the authors can state that they measure the particle depol ratio directly and not the volume depol ratio). I consider this as important, because what the authors present would be not valid for a lidar with similar technique but operating at a shorter wavelength.

25    Therefore I recommend, either to show the formula for the particle depol, indicating why you can neglect molecular contributions, or a proper referring to other publications. This starts already in the abstract, while you are stating the bleed trough is sufficient to measure particle depol.

*We have added more details on how attenuated backscatter and depolarization ratio are obtained from the Halo Doppler lidar in Sect. 2.1. We have also modified the abstract to highlight that for this lidar system, operating at a comparatively long wave-*
30  *length, instrumental noise floor and the internal polarizer performance are the main sources of uncertainty in depolarization ratio retrieval.*

*In the abstract:*

*It has been demonstrated that Halo Doppler lidars have the capability for retrieving the aerosol particle depolarization ratio at a wavelength of 1565 nm. For these lidars operating at such a long wavelength, the retrieval quality depends to a*
35  *large degree on an accurate representation of the instrumental noise floor and the performance of the internal polarizer, whose stability have not yet been assessed in long-term operation.*

*In Sect 2.1:*

*For a coherent Doppler lidar, attenuated backscatter ($\beta$') is calculated from SNR through*

$$\beta'(z) = A\frac{SNR(z)}{T_f(z)}, \tag{1}$$

40  *where A incorporates system-specific constants and $T_f(z)$ is the telescope focus function, which depends on range (z), the effective beam diameter and focal length of the system (Frehlich and Kavaya, 1991; Pentikäinen et al., 2020). For the Utö-32XR instrument, data from a co-located Vaisala CL31 ceilometer was utilised to determine the telescope function according to Pentikäinen et al. (2020) and then to calculate $\beta$' according to Eq. 1. For the non-XR instruments, $\beta$' was determined from the post-processed SNR using the 2 km focal length set in the firmware (Table 2).*

45  *Since A and $T_f(z)$ are equal for co- and cross-polar measurements, and attenuation can be assumed to be equal for both polarities, the depolarization ratio ($\delta$) can be retrieved as:*

$$\delta = \frac{SNR_{cross}}{SNR_{co}} \tag{2}$$

*At 1565 nm wavelength, the molecular backscatter coefficient is much smaller than at shorter wavelengths, being approx. $1.44 \times 10^{-8} m^{-1} sr^{-1}$ at standard pressure and temperature (Bucholtz, 1995). Additionally, atmospheric transmittance is very*
50  *close to unity in non-hydrometeor regions (Vakkari et al., 2021). Thus, we consider $\delta$ to be a fair estimate of the linear particle depolarization ratio without the correction for the molecular contribution (Vakkari et al., 2021). However, $\delta$ is sensitive to the performance of the internal polarizer as well as to the accuracy of the instrumental noise floor for both polarities (Vakkari et al., 2021).*

55    Furthermore, please make clear in the beginning what is the difference between depol_particle and depol_aerosol.

Indeed, there is no difference. We have decided to minimize the confusion by replacing $\delta_{aerosol}$ notation with "$\delta$ of aerosol", which is the particle depolarization ratio in aerosol region.

As for the depol ratio, it should be also briefly stated, why the attenuated backscatter can be used instead of the particle backscatter. No need for long formulas, but a proper referring or short introduction is needed, i.e., why the transmission is close to 1 at this wavelength and therefore the att. bsc. is close to particle bsc. in non-cloud regions.

Now that we are using a single lens system the only assumption that is needed here is that attenuation is equal for both polarities. This is now included in the modified Sect 2.1 (please see our response to the first general comment above).

In addition, please correct the unit for the attenuated backscatter throughout the manuscript ($\mathrm{Mm}^{-1}\,\mathrm{sr}^{-1}$ instead of $\mathrm{Mm}^{-1}$ only).

Thank you for spotting this, we have corrected the units throughout the manuscript.

The bleed-through is determined at cloud base, with a lot of efforts which are really appreciated (specially to avoid saturation). However, could you add a short statement, that the determined "B" is valid for the whole profile, i.e. also in conditions with much less SNR?

Thanks for the suggestion, we added this statement to line 156:
*Depending on cloud base height, the $SNR_{co}$ at cloud base can be as low as 0.01. As seen in Fig S1, there are no non-linear effects in the low $SNR_{co}$ end of the spectrum, and we have no reason to suspect that the polarizer performance would deteriorate with decreasing $SNR_{co}$. We also note that previous observations with Halo Doppler lidar report near-zero depolarization ratio for marine aerosol, which we expect to be spherical and therefore have very low depolarization ratio (Vakkari et al., 2021). Therefore, we are confident that bleed-through obtained from cloud-base observations can be used to correct also weak aerosol signals.*

Figure 2a: The depolarization ratios below the cloud base are significantly scattered, with values of -0-4 and + 0.07. Giving this plot, I could doubt the depolarization measurements in low SNR regimes. Can you comment on this?

The scatter in Fig. 1 (as Fig. 2a in the manuscript) illustrates the need for longer integration time for the weak aerosol signals, while in-cloud observations benefit from high time resolution. We modified the plot and add discussion to the manuscript.
*It should be noted that at the original integration time of 6s, the strong signal from cloud base results in minimal noise level of $\delta$ as seen in Fig. 2a. However, for the weak aerosol signal, $\delta$ at 6s integration time is rather noisy. Averaging over longer time periods reduces the instrumental noise in the aerosol $\delta$ substantially, as seen in the 1h averaged $\delta$ for aerosol part of the profile in Fig. 2a.*

[Figure]

**Figure 1 (as Figure 2a in the manuscript).** *Atmospheric profiles observed on 2018-09-24 at 05:35:17 UTC at Hyytiälä up to 1.4 km a.g.l. a) Depolarization ratio ($\delta$) at the original integration time of 6s and as 1h average for the aerosol part of the profile. Error bar indicates the measurement uncertainty. b) Signal-to-noise ratio in the co-polar channel ($SNR_{co}$). c) Attenuated backscatter ($\beta'$)*

Aerosol indicator description: As you refer many times to Figure S3, I would prefer to include in in the manuscript. On the other hand, you never refer to Fig. 3 a – c. Please find a compromise! I think it would be helpful, to use the plots 3a-c for explaining the typing methodology together with plot S3.

Thank you for the comment, we decided to simplify the description of the algorithm in the main part of the manuscript and move some of the technical details to the supplement. Updated Fig. 2 (as Fig. 3 in the manuscript) and text replacing lines 170-208 is below.

[Figure]

**Figure 2 (as Figure 3 in the manuscript).** *Atmospheric profiles on 12th of August 2018 at Hyytiälä. The left column displays the measured data where the background noise has been filtered for visualization in [a, c]. a) Attenuated backscatter (β'), c) Vertical velocity (w), e) Signal-to-noise ratio in the co-polar channel ($SNR_{co}$). The right column displays the steps from the Aerosol Identification algorithm. b) First step, d) Second step, f) Third step, g) Final step, i.e., the final result.*

95      *Figure 3a, 3c, 3e display the measured data from the Doppler lidar in 12th of August 2018 at Hyytiälä. The detail of the Aerosol Identification Algorithm is described in Sect. S3, a brief overview is explained as follows:*

*1. The first step of the algorithm involves detecting potential hydrometeors and aerosols from background signals based on β' and $SNR_{co}$. The result of this step is shown in Fig. 3b.*

*2. The falling hydrometeor detection step involves separating aerosol in downdrafts due to boundary layer mixing from actual*

100     *precipitation using both β' and w. Regions containing both up- and down- drafts are considered to be characteristic of boundary layer mixing, while regions of continuous downdrafts indicate precipitation. The result of this step is shown in Fig. 3d.*

*3. The attenuation correction step flagged all observations above clouds and precipitation with their corresponding class since the signal has been heavily attenuated. The result of this step is shown in Fig. 3f.*

*4. In the final step, a fine-tuned aerosol identification process is utilized to improve the aerosol class determination accuracy.*

*First, aerosol clusters are identified using both time and height domain. Then based on the average speed of the aerosol cluster and its connectiveness to the first lidar range gate, it can be classified as either aerosol, hydrometeor or undefined. The final result is shown in Fig. 3g.*

*The resulting classes are background signal, aerosol, hydrometeor, and undefined. For this algorithm, hydrometeors are defined as cloud (liquid or ice) or precipitation (rain or snow) and do not include aerosol.*

Post processing: Is mainly very technical and hard to understand as sometimes further information is missing. Is it possible to have a more detailed part of the description in the appendix and shorten the technical part in the manuscript? For example, move lines 212 to 229 to the appendix and extent the description a bit, so that other groups with HALO lidars can use similar approaches.

We agreed to your comment and have moved the technical part to the supplement. We also simplified the post processing part following your comment and the other referee's.

In the main text:

*Aerosol is expected to be well-mixed within each aerosol layer, so in order to extract weak aerosol signal and minimize the random noise, $SNR_{co}$ and $SNR_{cross}$ were averaged for 1 hour.*

*As mentioned before, the SNR data in this study have been processed with the background correction algorithm described by Vakkari et al. (2019). Briefly, the noise floor consists of a non-polynomial component, which is obtained from the background checks according to Vakkari et al. (2019) and a polynomial component, which is obtained from a fit to the aerosol- and hydrometeor- free (background) range gates of each $SNR_{co}$ and $SNR_{cross}$ profile (Manninen et al., 2016; Vakkari et al., 2019). Typically, the linear part of the noise floor is much larger than the 2nd order polynomial component, but for extended averaging (more than 1 hour) it is essential to include in the background correction. An example of this is shown in Fig. 4a demonstrating how this 2nd order polynomial component can greatly affect the $\delta$ of aerosol retrieval in aerosol layers with low SNR (Fig. 4d). Previously (Vakkari et al., 2021; Bohlmann et al., 2021), the 2nd order component of the noise floor has been fitted to aerosol- and hydrometeor- free range gates of the SNR profiles based on visual inspection of individual profiles. However, given the large number of profiles analysed in this study, this approach is not feasible and thus we have automated the fitting of the 2nd order polynomial. The fitting algorithm is described in detail in the Sect. S2, and the resulting $SNR_{co}$ and $SNR_{cross}$ profiles are shown in Fig. 4b, 4c.*

*The attenuated backscatter is calculated from the background-corrected $SNR_{co}$. Next, aerosol layer(s) are identified using the Aerosol Identification algorithm. Finally, following Vakkari et al. (2021), the bleed-through corrected $\delta$ in aerosol regions is calculated as*

$$\delta = \frac{SNR_{cross} - B \cdot SNR_{co}}{SNR_{co}} \tag{3}$$

*where B is the estimated bleed-through of each instrument (Table 3).*

*The resulting $\delta$ is shown in Fig. 4d, and the estimation of its uncertainty is presented in Sect. S2. Additionally, the post-processed*

*δ of aerosol was collected from the whole dataset and compared with the original δ. The result described in Sect. S2 shows that the post-processing procedure substantially improved the δ of aerosol with low SNR values.*

140   In the supplement:

*For a successful fit, aerosol- and hydrometeor- free (background or noise-only) range gates need to be identified in the profile. Firstly, the data is averaged every hour, and the $SNR_{co}$ and $SNR_{cross}$ profiles are then decomposed by stationary wavelet transform (Nason and Silverman, 1995) with the wavelet bior2.6 using Pywavelets (Lee et al., 2019). Next, the variance of the noise in the SNR is removed by applying a hard threshold shrinkage function using universal thresholding (Donoho*
145   *and Johnstone, 1994) to the approximation and detail coefficients from level 1 to level 4 resulted from the wavelet transform (Nason and Silverman, 1995). The SNR is then reconstructed using inverse stationary wavelet transform (Nason and Silverman, 1995). Finally, the background range gates are identified as those having reconstructed $SNR_{co}$ values less than the standard deviation of the instrument noise floor divided by the squared root of the number of profiles averaged in that hour. The 2nd order polynomial fit can then be performed on those noise-only data points identified in the profile. The fit follows Eq. (S1) and*
150   *the correction is then applied to the entire SNR profile, similar to (Vakkari et al., 2019).*

$$SNR(z) = a + h_z \cdot b + c \cdot h_z^2 \tag{S1}$$

*where $SNR_z$ and $h_z$ are the background SNR and height at each range gate, z, and a,b,c are the parameters of the fit for each profile.*

*Next, the aerosol-only range gates are identified based on the result of the Aerosol Identification algorithm applied to the orig-*
155   *inal non-averaged data. If 80 percent of original the non-averaged measurements at a range gate in an hour were identified as aerosol by the Aerosol Identification algorithm, then that 1-hour averaged range gate is identified as aerosol measurement. Finally, following Vakkari et al. (2021), the bleed-through corrected δ of aerosol can be calculated following Eq. (2), utilizing the estimated bleed through obtained in Table 3.*

160   Specific:

line 43: Seems to be wrong reference for narrow Calipso swath?

Thanks for spotting this, this is corrected as:

*Space-borne lidars such as CALIPSO (Cloud-Aerosol Lidar and Infrared Pathfinder Satellite Observation; Winker et al. (2009)) cover the globe but with low temporal and spatial resolution due to their very narrow swath.*

165

Line 46: Please rephrase sentence.

Rephrased as:

*These lidar networks enable the monitoring in real-time of the aerosol vertical profiles in different environments across a large aera. Consequently, they facilitate the detection of elevated aerosol layers and the investigation of long-term vertical atmo-*

*spheric properties.*

108: Remove brackets from citation

Done

109: Why do you need the focal length to determine the attenuated backscatter, please briefly explain or give proper reference.

The formula for the attenuated backscatter $\beta'$ is described in Eq. 1. in the new modified Sect. 2.1 including references. We hope this explains the matter.

*For a coherent Doppler lidar, attenuated backscatter ($\beta$') is calculated from SNR through*

$$\beta'(z) = A\frac{SNR(z)}{T_f(z)} \tag{1}$$

*, where A incorporates system-specific constants and $T_f(z)$ is telescope focus function, which depends on range (z), the effective beam diameter and the focal length of the system (Frehlich and Kavaya, 1991; Pentikäinen et al., 2020)*

166: I find this abbreviation "AI" quite challenging, giving the fact that it is often used for artificial intelligence. Consider to change to AIA?

We will remove the use of this abbreviation and replace it with its full name "Aerosol Identification algorithm"

178: Here you use a threshold of 10^-5.5 for aerosol cloud discrimination. Later you use 10^-7 for precipitation. This is a bit in contradiction. Can you explain better?

The first step of the algorithm performs a preliminary guess that aerosol has $\beta' < 10^{-5.5}$ and hydrometeor (cloud + precipitation) has $\beta' > 10^{-5.5}$. This is a preliminary guess because light precipitation is difficult to be separated from aerosol using just $\beta'$. In the falling hydrometeor step, $w$ was utilized to detect falling hydrometeors from both aerosol and hydrometeor mask above. This falling hydrometeor mask is then applied back to and overwrite the preliminary guess of aerosol and hydrometeor. We could have categorized this "falling hydrometeor" mask as a new category "precipitation", but this falls beyond the scope of this study so we decided not to. The threshold of $10^{-7}$ in $\beta'$ was used in the falling hydrometeor detection step to discard the aerosol downdraft, with the assumption that falling hydrometeor always has $\beta' > 10^{-7}$.
We added the following in Sect. S3:

*The precipitation data points are then overwritten the aerosol data points from step 1 as hydrometeor.*

179: Please also briefly describe what you mean with a median and maximum kernel. It is yet ambiguous.

We adopt the use of 2D-kernel from image processing methods. This is now clarified:

*The Aerosol Identification algorithm developed here utilizes 2D-kernel manipulation, which is a commonly used approach in image processing (Guo et al., 2022; Li et al., 2013; Perreault and Hébert, 2007), to extract various features from the data and to determine the correct class for each data point. A kernel, also referred to as a filter, template, window or mask (Gonzalez and Woods, 2007) is a small 2-D data array. Mathematical operations, such as median, maximum, Gaussian etc., on all values inside the kernel determine its center value. The kernel is run through each data point one by one, replacing its center value with mathematical operations of the neighboring values.*

210: "....were averaged for 1 hour before calculating delta_aerosol." So basically delta_aerosol is just the average of delta_particle in aerosol only regions, correct? Please write exactly what you mean.

Yes, $SNR_{co}$ and $SNR_{cross}$ were averaged for 1 hour. Then $\delta$ were calculated based on these averages following Eq. 3. As mentioned previously, we also replaced $\delta_{aerosol}$ by $\delta$ of aerosol to minimize the confusion.

213: "...a weak 2nd order polynomial shape (c.f. Manninen et al., 2016) appears in SNRco and SNRcross..." : From the given reference, it is not clear what is meant with second order polynomial shape appearing in SNR. Please describe in more detail. Does it mean electronic oscillations in the signal? It becomes more clear later, but should be clearly described here.

We have added more details and simplified this post-processing section. Please see our response to the previous comment for the full post-processing section. Here is our modified paragraph:

*As mentioned before, the SNR data in this study have been processed with the background correction algorithm described by Vakkari et al. (2019). Briefly, the noise floor consists of a non-polynomial component, which is obtained from the background checks according to Vakkari et al. (2019) and a polynomial component, which is obtained from a fit to the aerosol- and hydrometeor- free (background) range gates of each $SNR_{co}$ and $SNR_{cross}$ profile (Manninen et al., 2016; Vakkari et al., 2019). Typically, the linear part of the noise floor is much larger than the 2nd order polynomial component, but for extended averaging (more than 1 hour) it is essential to include in the background correction. An example of this is shown in Fig. 4a demonstrating how this 2nd order polynomial component can greatly affect the $\delta$ of aerosol retrieval in aerosol layers with low SNR (Fig. 4d). Previously (Vakkari et al., 2021; Bohlmann et al., 2021), the 2nd order component of the noise floor has been fitted to aerosol- and hydrometeor- free range gates of the SNR profiles based on visual inspection of individual profiles. However, given the large number of profiles analysed in this study, this approach is not feasible and thus we have automated the fitting of the 2nd order polynomial. The fitting algorithm is described in detail in the Sect. S2, and the resulting $SNR_{co}$ and $SNR_{cross}$ profiles are shown in Fig. 4b, 4c.*

215: I do not understand this part of the sentence: "...2021), this component of the noise floor has been accounted for through a fit to SNR profiles, ..." Please rephrase!

This has been rephrased. Please see our response to the previous comment for the modified text.

220: What is "with wavelet bior2.6." ??? is this a fragment or a reference missing?

We have added a reference for this wavelet and moved the text to the supplement:

*Firstly, the data is averaged every hour, and the $SNR_{co}$ and $SNR_{cross}$ profiles are then decomposed by stationary wavelet transform (Nason and Silverman, 1995) with the wavelet bior2.6 using Pywavelets (Lee et al., 2019)*

240

221: "Next, the variance of the noise in the SNR is removed by applying a hard threshold shrinkage function using universal thresholding (Donoho and Johnstone, 1994) to the approximation and detail coefficients from level 1 to 4." Please explain in more detail. This is not understandable. For example, what is level 1 to 4? Was never introduced.

This technical part is moved to the supplement and expanded as:

245 *Next, the variance of the noise in the SNR is removed by applying a hard threshold shrinkage function using universal thresholding (Donoho and Johnstone, 1994) to the approximation and detail coefficients from level 1 to level 4 resulting from the wavelet transform (Nason and Silverman, 1995)*

Line 234: A reference to table 3 would be already great here.

250 We have added a reference to Table 3 as suggested.

Figure 4d: Please enlarge the scale of x-axis so that one can see the depol values in the ice cloud (is it an ice cloud?). I.e. from 0 to 0.6

We followed your suggestion and extended the x-axis. The high depol values at 8 km are from ice clouds.

255

Figure 4b-d are never referenced. Please do so at the appropriate places in the text.

We have now referred to them in the text

*An example of this is shown in Fig. 4a demonstrating how this 2nd order polynomial component can greatly affect the $\delta$ of aerosol retrieval in aerosol layers with low SNR (Fig. 4d).*

260 *The fitting algorithm is described in detail in the Sect. S2, and the resulting $SNR_{co}$ and $SNR_{cross}$ profiles are shown in Fig. 4b, 4c.*

244: 2.3.1 subsection is introduced, but there is no 2.3.2 Please check sectioning in whole manuscript.

Thank you for spotting this. We moved the Sect 2.3.1 to the supplement

265

261: Sentence not understandable without further background. What "chains" shall converge? What are sampling chains? Please expand description here (or more in appendix)

This technical bit is moved to the supplement:

*Four different sampling chains (van Ravenzwaaij et al., 2018) are utilized with 2000 samples for each chain (excluding the*
*500 warmup/burn-in samples), and convergence diagnostics have been carried out to ensure that the chains converged*

421: Any idea what kind of aerosol this can be in September with high depol? I consider for Pollen it's too late, is it?

Indeed, it is likely too late for pollen. At this wavelength, so far only pollen and mineral dust have been observed to have such high $\delta$. However, we haven't investigated the origins of these layers in more detail, and prefer to leave it for further studies to limit the length of this manuscript.

431: The minimum range of the lidar is stated later, but it would be good to give the values already here and maybe even in Tab. 2.

Minimum range is added to Table 2 and the sentence here is changed as:
*above the lidar's minimum range of 90 m (Table 2).*

438: 3.4.1 subsection is introduced, but there is no 3.4.2 Please check sectioning in whole manuscript.

Thanks for spotting this, we have now fixed the numbering

465: The case studies come a bit out of nothing, but are very interesting. However, please consider to move them to before the long-term analysis....Furthermore, do you have one case study with low depol in summer?

We took your suggestion and moved the case studies before the long-term analysis. There are cases of low $\delta$ in summer, but the manuscript is already rather long and we decided to leave them out from this study. However, we expanded the case study on the long-range transported mineral dust and the updated figure now displays also some profiles with low $\delta$. Please see our next response for the updated figures.

472: To compare the delta_aerosol to dust values at other wavelengths is a bit critical. Especially, as you later show similar values for Pollen. Furthermore, some studies show that depol ratio for dust is highest in the visible and is decreasing towards longer wavelengths (1064 nm). Can you comment on this? Furthermore, do you have any more information/proof for dust presence in this case study? E.g. DREAM or CAMS model prediction? Evidence from nearby lidar stations?

Compared to e.g. smoke, the wavelength dependency of dust $\delta$ is moderate, as seen in Vakkari et al. (2021). On the other hand, pollen and dust $\delta$ cover more or less the same range at 1565nm (Vakkari et al., 2021; Bohlmann et al., 2021). That is, elevated $\delta$ at 1565 nm alone is not enough to identify a layer as dust or pollen. However, this case was investigated in more detail by Lobo (2021) and the air mass origin together with CALIOP data (Fig. 3 (as Figure S3 in the manuscript)) offer strong evidence of Saharan dust for the layer in question. In addition, CAMS indicates the presence of dust over Finland at the time

of this case study (Fig. 4 (as Figure S4 in the manuscript)). We do not have concurrent measurements from lidars at other wavelengths for this layer, but we noticed that the same elevated layer is observed by the Halo Doppler lidar at Utö. We have included the profiles from Utö lidar in the main part of the manuscript and the CALIOP and CAMS plots in the supplement. We have removed the reference to observations at shorter wavelengths from the discussion and modified the text in the main
305 part of the manuscript as:

*From late night 14th of April 2018 to morning 15th of April 2018, an elevated aerosol layer between 2 km to 4 km a.g.l. was observed above Hyytiälä (Fig. 8a, 8b). The average profiles (Fig. 8e, 8f) from 03:00 to 04:00 UTC on 15th of April 2018 show an elevated aerosol layer with the $\delta$ of aerosol at 0.24 $\pm$ 0.008, while in the boundary layer (from the surface to 1.5 km a.g.l.),*
310 *the $\delta$ of aerosol is at 0.12 $\pm$ 0.004. Similarly, an elevated aerosol layer between 2 km to 3 km was also observed in Utö (Fig. 8c, 8d). The averaged profiles (Fig. 8e, 8f) from 00:00 to 01:00 UTC on 15th of April show that the $\delta$ of this layer is at 0.226 $\pm$ 0.005 while in the boundary layer (from the surface up to 2km), the $\delta$ of aerosol is between 0.05 and 0.1. These substantial difference in the $\delta$ of aerosol between the elevated layer and the boundary layer indicates that the elevated aerosol had likely undergone long-range transport to these sites.*

315 *The FLEXPART simulation of this layer (Fig. S5) indicates that the air mass was not in contact with the surface for 4.5 days prior to arrival over Hyytiälä. However, 5-7 days before arrival, a contribution from the surface layer was seen coming mostly from the western Sahara, with minor contributions from the Mediterranean. On 15th April an overpass by CALIPSO crossed the air mass transport pathway relatively close to the observation sites in Finland at 62.8N,24.3E and the CALIOP aerosol subtype V4.2 product indicates a dust layer at 3-5 km above ground at this location (Fig. S3). On 15 April also dust aerosol*
320 *optical depth at 550nm from the CAMS model (Benedetti et al., 2009; Morcrette et al., 2009) indicates presence of dust over south western Finland (Fig. S4).*
*These results suggest the layer to be most likely Saharan dust, although the observed $\delta$ is slightly lower than the value of 0.30 reported by Vakkari et al. (2021) for dust at 1565 nm. This could be due to gravitational settling of large dust particles during long range transport (Haarig et al., 2017).*

[Figure]

**Figure 3 (as Figure S3 in the manuscript).** *Aerosol subtype V4.2 (Kim et al., 2018; Liu et al., 2019) derived from CALIOP data onboard CALIPSO Winker et al. (2009) on 2018-04-15. A layer of dust can be observed near Hyytiälä lidar ($61.84^o N$, $24.29^o E$)*

[Figure]

**Figure 4 (as Figure S4 in the manuscript).** *Dust Aerosol Optical Depth at 550nm from CAMS model forecast (Benedetti et al., 2009; Morcrette et al., 2009)*

[Figure]

**Figure 5 (as Figure 8 in the manuscript).** *Profiles from 2018-04-14 to 2018-04-15 in Hyytiälä and Utö. a) Attenuated backscatter (β') in Hyytiälä, b) 1-hour averaged depolarization ratio (δ) in Hyytiälä. c) Attenuated backscatter (β') in Utö, d) 1-hour averaged depolarization ratio (δ) in Utö. And 300m running mean (in shaded area) on 2018-04-15 03:00-04:00 UTC in Hyytiälä and 2018-04-15 00:00-01:00 UTC in Utö of c) β' profiles c) δ profiles.*

325 This also reminds me, that it would be interesting if you ever have observed a lofted aerosol layer in summer with low depol? Smoke or so? Or is your depol always high?

Fig. 6 shows an elevated aerosol layer in the summer with a low $\delta$ of 0.05 on top of the high $\delta$ surface aerosol. This layer could be smoke, but to limit the length of the manuscript we leave more detailed analysis of this case for a further study.

[Figure]

**Figure 6.** *Profiles on 2018-05-09 at Hyytiälä a) Attenuated backscatter ($\beta$'), b) 1-hour averaged depolarization ratio ($\delta$). And 300m running mean (in shaded area) on 2018-05-09 14:00-15:00 UTC of c) $\beta$' profiles c) $\delta_{aerosol}$ profiles*

330 500: Please also state here already somewhere that you developed a complete new AIA for HALO systems. This is really remarkable.

We have added the following sentence on line 500:
*In order to facilitate the second aim, an Aerosol Identification algorithm was created utilizing only data from Halo Doppler lidars.*

335

515: "The overall delta_aerosol are 0.13 ± 0.08 in Hyytiälä, 0.11 ± 0.07 in Vehmasmäki, 0.09 ± 0.07 in Utö and 0.07 ± 0.08 in Sodankylä.": Interesting! This means that in contrast to many other places in the world, you are mostly prone to Pollen (continental background) in summer in case of clear sky. Maybe worth stating

Thank you for the recommendation, we have now emphasized the presence of pollen in the Finnish clean background air.

340 *From the four-year observations of the $\delta$ of aerosol, we found that it is surprisingly similar at all four sites in Finland. The*

*overall δ of aerosol are 0.13 ± 0.08 in Hyytiälä, 0.11 ± 0.07 in Vehmasmäki, 0.09 ± 0.07 in Utö and 0.07 ± 0.08 in Sodankylä. All these sites have low value of δ of aerosol in the winter months and higher value in the summer months, which is attributed to the presence of irregular-shaped pollen particles in relatively clean background air.*

345      542: Code availability: It would be great if you could make a release on github and publish this frozen version (referred to in this publication) on zenodo or similar for getting an DOI.

The code is available on github and a frozen version published with a DOI, please see the next response.

545: Data availability: Lidar data are available upon request from the authors. Such statements are not anymore acceptable.
350   Please consider to provide the data via a data portal.

Thank you for your comment, we completely agree with your feedback. We are in the process of uploading the data and the code to our data portal. Some of the following DOIs will be available for access soon.
We have added the following text to:

Data availability
355   *The data used in this study are generated by the Aerosol, Clouds and Trace Gases Research Infrastructure (ACTRIS) and are available from the ACTRIS Data Centre using the following link:*
*https://doi.org/10.60656/919d6e2a0e454c18.*
*https://doi.org/10.23728/fmi-b2share.f82603e69cea49b888f94d0e8a85e787.*

Acknowledgements
360   *We acknowledge ACTRIS and Finnish Meteorological Institute for providing the Cloudnet classification data set which is available for download from https://cloudnet.fmi.fi. We acknowledge ECMWF for providing ERA5 model reanalysis data, and DWD for providing ICON model data.*

Citation
*Moisseev, D., O'Connor, E., and Petäjä, T. (2023). Custom collection of Cloudnet classification data from Hyytiälä between 26*
365   *Nov 2016 and 31 Dec 2019. ACTRIS Cloud remote sensing data centre unit (CLU).*
*https://doi.org/10.60656/919d6e2a0e454c18*
*Le, V. (2023). Data and code for "Long-term aerosol particle depolarization ratio measurements with Halo Doppler lidar" by Viet Le et al. (2023), B2SHARE [data and code]*
*https://doi.org/10.23728/fmi-b2share.f82603e69cea49b888f94d0e8a85e787*
370

Supplement:
Caption Table S2 need to be expanded: More information needed! What is shown? The increase in data points? Not clear now.

We modified the caption

375 *Percentage of aerosol data with the changes of δ resulted from the post-processing procedure. The second column displays the percentage of aerosol data that has $SNR_{co}$ changed more than 0.05. The third column displays the percentage of aerosol data that has δ changed more than 0.1.*

We added clarification to the text

*Fig. S2 demonstrates the impact of the background correction to δ of aerosol. For all the instruments except Sodankylä-*
380 *54, the background correction increases δ of aerosol. The increase is higher for aerosol with weaker $SNR_{co}$. The effect is negligible for strong aerosol signal with $SNR_{co}$ larger than 0.01. Table S1 shows the percentage of data affected by this background correction. The effect is most prominent for the instruments at Hyytiälä; as 31.8% and 24.6% of the aerosol data have δ changed by 0.05 for Hyytiälä-33 and Hyytiälä-46 respectively. However, significant changes of aerosol δ more than 0.1 is only at 9.3% and 7.1% for these instruments. This result shows the importance of background correction in the retrieval of*
385 *weak aerosol signals. Without it, biases from the 2nd order polynomial component in the background would propagate into the δ of aerosol*

Caption Table S4 need to be expanded: More information needed: Linear regression between what? I.e. what is y, what is x? What about an offset? What is p what is R squared?

390 *Linear regression analysis summary for δ = Slope × RH + off-set (not shown in the table), where δ is the δ of aerosol and RH is the surface relative humidity (at 2 m a.g.l). The p-values for the slopes are < 10-3, indicating that all the slopes' values are statistically significant (i.e., different than 0). The R squared describes the proportion of variance in δ that can be explained by RH*

395 Please check all captions in supplement!

Done.

**References**

Benedetti, A., Morcrette, J.-J., Boucher, O., Dethof, A., Engelen, R. J., Fisher, M., Flentje, H., Huneeus, N., Jones, L., Kaiser, J. W., Kinne, S., Mangold, A., Razinger, M., Simmons, A. J., and Suttie, M.: Aerosol analysis and forecast in the European Centre for Medium-Range Weather Forecasts Integrated Forecast System: 2. Data assimilation, Journal of Geophysical Research: Atmospheres, 114, https://doi.org/10.1029/2008JD011115, _eprint: https://onlinelibrary.wiley.com/doi/pdf/10.1029/2008JD011115, 2009.

Bohlmann, S., Shang, X., Vakkari, V., Giannakaki, E., Leskinen, A., Lehtinen, K. E., Pätsi, S., and Komppula, M.: Lidar depolarization ratio of atmospheric pollen at multiple wavelengths, Atmospheric Chemistry and Physics, 21, https://doi.org/10.5194/acp-21-7083-2021, 2021.

Bucholtz, A.: Rayleigh-scattering calculations for the terrestrial atmosphere, Applied Optics, 34, 2765–2773, https://doi.org/10.1364/AO.34.002765, publisher: Optica Publishing Group, 1995.

Donoho, D. L. and Johnstone, J. M.: Ideal spatial adaptation by wavelet shrinkage, Biometrika, 81, https://doi.org/10.1093/biomet/81.3.425, 1994.

Frehlich, R. G. and Kavaya, M. J.: Coherent laser radar performance for general atmospheric refractive turbulence, Applied Optics, 30, 5325–5352, https://doi.org/10.1364/AO.30.005325, publisher: Optica Publishing Group, 1991.

Gonzalez, R. C. and Woods, R. E.: Digital Image Processing (3rd Edition), publication Title: Prentice-Hall, Inc. Upper Saddle River, NJ, USA ©2006, 2007.

Guo, S., Wang, G., Han, L., Song, X., and Yang, W.: COVID-19 CT image denoising algorithm based on adaptive threshold and optimized weighted median filter, Biomedical Signal Processing and Control, 75, https://doi.org/10.1016/j.bspc.2022.103552, 2022.

Haarig, M., Ansmann, A., Althausen, D., Klepel, A., Groß, S., Freudenthaler, V., Toledano, C., Mamouri, R. E., Farrell, D. A., Prescod, D. A., Marinou, E., Burton, S. P., Gasteiger, J., Engelmann, R., and Baars, H.: Triple-wavelength depolarization-ratio profiling of Saharan dust over Barbados during SALTRACE in 2013 and 2014, Atmospheric Chemistry and Physics, 17, 10 767–10 794, https://doi.org/10.5194/acp-17-10767-2017, 2017.

Kim, M. H., Omar, A. H., Tackett, J. L., Vaughan, M. A., Winker, D. M., Trepte, C. R., Hu, Y., Liu, Z., Poole, L. R., Pitts, M. C., Kar, J., and Magill, B. E.: The CALIPSO version 4 automated aerosol classification and lidar ratio selection algorithm, Atmospheric Measurement Techniques, 11, https://doi.org/10.5194/amt-11-6107-2018, 2018.

Lee, G., Gommers, R., Waselewski, F., Wohlfahrt, K., and O'Leary, A.: PyWavelets: A Python package for wavelet analysis, Journal of Open Source Software, 4, https://doi.org/10.21105/joss.01237, 2019.

Li, S., Kang, X., and Hu, J.: Image fusion with guided filtering, IEEE Transactions on Image Processing, 22, https://doi.org/10.1109/TIP.2013.2244222, 2013.

Liu, Z., Kar, J., Zeng, S., Tackett, J., Vaughan, M., Avery, M., Pelon, J., Getzewich, B., Lee, K. P., Magill, B., Omar, A., Lucker, P., Trepte, C., and Winker, D.: Discriminating between clouds and aerosols in the CALIOP version 4.1 data products, Atmospheric Measurement Techniques, 12, https://doi.org/10.5194/amt-12-703-2019, 2019.

Lobo, H.: Using the Finnish Doppler lidar network to study elevated aerosol depolariation ratio at 1565 nm (MSc Thesis), University of Helsinki, https://helda.helsinki.fi/items/688fa5a5-a7b3-418a-b760-641bdce5285a, 2021.

Manninen, A. J., O'Connor, E. J., Vakkari, V., and Petäjä, T.: A generalised background correction algorithm for a Halo Doppler lidar and its application to data from Finland, Atmospheric Measurement Techniques, 9, 817–827, https://doi.org/10.5194/amt-9-817-2016, 2016.

Morcrette, J.-J., Boucher, O., Jones, L., Salmond, D., Bechtold, P., Beljaars, A., Benedetti, A., Bonet, A., Kaiser, J. W., Razinger, M., Schulz, M., Serrar, S., Simmons, A. J., Sofiev, M., Suttie, M., Tompkins, A. M., and Untch, A.: Aerosol analysis and forecast in the

435    European Centre for Medium-Range Weather Forecasts Integrated Forecast System: Forward modeling, Journal of Geophysical Research: Atmospheres, 114, https://doi.org/10.1029/2008JD011235, 2009.

Nason, G. P. and Silverman, B. W.: The Stationary Wavelet Transform and some Statistical Applications, https://doi.org/10.1007/978-1-4612-2544-7_17, 1995.

Pentikäinen, P., James O'connor, E., Juhani Manninen, A., and Ortiz-Amezcua, P.: Methodology for deriving the telescope focus function and
440    its uncertainty for a heterodyne pulsed Doppler lidar, Atmospheric Measurement Techniques, 13, 2849–2863, https://doi.org/10.5194/amt-13-2849-2020, 2020.

Perreault, S. and Hébert, P.: Median filtering in constant time, IEEE Transactions on Image Processing, 16, https://doi.org/10.1109/TIP.2007.902329, 2007.

Vakkari, V., Manninen, A. J., O'Connor, E. J., Schween, J. H., Van Zyl, P. G., and Marinou, E.: A novel post-processing algorithm for Halo
445    Doppler lidars, Atmospheric Measurement Techniques, 12, 839–852, https://doi.org/10.5194/amt-12-839-2019, 2019.

Vakkari, V., Baars, H., Bohlmann, S., Bühl, J., Komppula, M., Mamouri, R. E., and O'connor, E. J.: Aerosol particle depolarization ratio at 1565 nm measured with a Halo Doppler lidar, Atmospheric Chemistry and Physics, 21, https://doi.org/10.5194/acp-21-5807-2021, 2021.

van Ravenzwaaij, D., Cassey, P., and Brown, S. D.: A simple introduction to Markov Chain Monte–Carlo sampling, Psychonomic Bulletin & Review, 25, 143–154, https://doi.org/10.3758/s13423-016-1015-8, 2018.

450    Winker, D. M., Vaughan, M. A., Omar, A., Hu, Y., Powell, K. A., Liu, Z., Hunt, W. H., and Young, S. A.: Overview of the CALIPSO Mission and CALIOP Data Processing Algorithms, Journal of Atmospheric and Oceanic Technology, 26, 2310–2323, https://doi.org/10.1175/2009JTECHA1281.1, 2009.

---

## Author Comment (AC2)

**Response to reviewers comments for AMT-2023-37**

Viet Le[1], Hannah Lobo[1], Ewan J. O'Connor[1], and Ville Vakkari[1, 2]

[1]Finnish Meteorological Institute, Helsinki, 00101, Finland
[2]Atmospheric Chemistry Research Group, Chemical Resource Beneficiation, North-West University, Potchefstroom, 2520, South Africa

**Correspondence:** viet.le@fmi.fi

We would like to thank the reviewers for the constructive comments. Below are the reviewer comments in black text, followed by our responses in blue text. Where applicable, we also provide changes to the manuscript in *blue italics*.

**1 Reviewer 2**

The manuscript demonstrates the capability of Halo doppler lidars to retrieve the particle linear depolarization ratio ($\delta$aerosol)
at 1565 nm and investigates the stability of the background noise levels and the performance of the internal polarizer, both of which have to be considered for the retrieval of $\delta$aerosol. The retrieved particle linear depolarization ratio is used to perform a seasonal characterization of the suspended aerosols at four different sites of Finland of different environments (e.g. marine, forest, rural, sub artic regions) using four-year lidar measurements from 2016 to 2019. For the seasonal analysis on the $\delta$aerosol, a new aerosol identification (AI) algorithm has been developed to separate the aerosol from the cloud layers. The performance
of the AI algorithm has been compared against the Cloudnet classification algorithm showing adequate performance on identifying and separating the aerosol layers from the clouds. Moreover, two case studies, one Saharan dust transport event and one pollen from continental areas event, have been selected to be presented in the study while using complimentary data for the air mass origin from FLEXPART model.

To sum up, the manuscript presents new methods and new developments falling within the scope of the journal. It is well-
structured and well-written even though I have the feeling that some parts could be further explained and/or discussed in order to be easier to follow for a reader non-relevant to the study. The scientific significance makes the manuscript suitable for publication in AMT, after some minor revisions have been considered from the authors.

We would like to thank the reviewer for their positive evaluation of the manuscript.

Comments:

A general comment for the authors is to explain better the difference between $\delta$ (is it the volume linear depolarization ratio?) and $\delta$_aerosol (particle linear depol. ratio?) in the text as it might be confusing for the readers

$\delta_{aerosol}$ is an abbreviation for $\delta$ of aerosol. To alleviate the confusion, we will replace $\delta_{aerosol}$ with $\delta$ of aerosol

Line 53: "...$\delta$_aerosol is measured using Raman lidar (Engelmann et al., 2016; Baars et al., 2016)"

The depolarization ratio profile can only be retrieved when a lidar is equipped with depolarization channels, thus from a depolarization lidar (not all Raman lidars have depolarization channels). PollyXTs are Raman and depolarization (and water-vapor) lidar systems. Please rephrase and add also older studies since depolarization in lidar measurements is being used even before 2016 (see for example Sassen, K.: Polarization in Lidar, in Lidar: Range-Resolved Optical Remote Sensing of the Atmosphere, edited by C. Weitkamp, pp. 19–42, Springer New York, New York, NY., 2005)

*We have rephrased this as:*

*Typically, $\delta$ of aerosol is measured at shorter wavelenghts such as at $355\mu m$, $0.523\mu m$, $0.532\mu m$, $0.694\mu m$, $0.710\mu m$ or $1.064\mu m$ (Murayama et al., 2001; Sassen, 2002; Engelmann et al., 2016; Baars et al., 2016)*

Line 124: "…estimate the internal polarizer performance, or bleed-through (Vakkari et al., 2021)". Here you could add a
short description about what the bleed-through is, in order to be easier for an independent reader to follow. Then if the reader wants more details, can read the Vakkari et al., 2021.

*The bleed-through is defined by Vakkari et al. (2021) as the incomplete extinction in the lidar internal polarizer, which the co-polar signals is leaking into the cross-receiver. This results in a systematic bias in the calculated $\delta$ from $SNR_{co}$ and $SNR_{cross}$.*

Line 147: "… The mean and standard deviation of $\delta$ at the cloud base were then calculated and used to determine the bleed through". Here the authors could also mention that the B term they are using in eq. 3 is the calculated mean value of $\delta$. Moreover, the values of $\delta$ at the cloud base do they correspond to an averaged $\delta$ inside the range gate that is identified as the cloud base?

*Thank you for the suggestion, we have now added to Sect 2.1.2:*
*The mean and standard deviation of $\delta$ at all the cloud bases were then calculated and used to determine the bleed-through for each instrument and to investigate its stability over time.*
*And to Sect 2.3:*
*where B is the estimated bleed-through of each instrument (Table 3).*
The values of $\delta$ at the cloud base is collected in each individual cloud profile of each instrument. The bleed-through $B$ is the
averaged $\delta$ of all these profiles of each instrument excluding the tail of the distribution due to mixed-phase cloud or non-ideal sampling of liquid clouds as described in Section 3.1.2.

Lines 159 – 263: These sections are too technical and refer to readers with specific expertise in HALO wind lidar technologies and data handling. Therefore, I would suggest to move these sections in an appendix and maybe lines 170-208 where the
steps of AI are described to be moved at the supplement below figure S3. Instead of this too technical description you could add a more qualitative description of the AI and the signal post-processing sections to achieve a smoother transition to section for the investigation of the air mass origins. However, Figure 3 is a good way to demonstrate the results on the target classification from AI and thus could remain in the main text but with further discussion.

Thank you for the comment, we decided to simplify the description of the algorithm in the main part of the manuscript and
move some of the technical details to the supplement. Updated Fig. 3 in the manuscript and text replacing lines 170-208 is
below.

[Figure]

**Figure 3.** *Atmospheric profiles on 12th of August 2018 at Hyytiälä. The left column displays the measured data where the background noise has been filtered for visualization in [a, c]. a) Attenuated backscatter ($\beta$'), c) Vertical velocity (w), e) Signal-to-noise ratio in the co-polar channel ($SNR_{co}$). The right column displays the steps from the Aerosol Identification algorithm. b) First step, d) Second step, f) Third step, g) Final step, i.e., the final result.*

*Figure 3a, 3c, 3e display the measured data from the Doppler lidar in 12th of August 2018 at Hyytiälä. The detail of the Aerosol Identification Algorithm is described in Sect. S3, a brief overview is explained as follows:*

*1. The first step of the algorithm involves detecting potential hydrometeors and aerosols from background signals based on $\beta$'*
*and SNRco. The result of this step is shown in Fig. 3b.*
*2. The falling hydrometeor detection step involves separating aerosol in downdrafts due to boundary layer mixing from actual precipitation using both $\beta$' and w. Regions containing both up- and down- drafts are considered to be characteristic of bound-*

*ary layer mixing, while a region of continuous downdrafts indicates precipitation. The result of this step is shown in Fig. 3d.*

*3. Attenuation correction step flagged all observations above clouds and precipitation with their corresponding class since the*
*signal has been heavily attenuated. The result of this step is shown in Fig. 3f.*

*4. In the final step, a fine-tuned aerosol identification process is utilized to improve the aerosol class determination accuracy. First, aerosol clusters are identified using both time and height domain. Then based on the average speed of the aerosol cluster and its connectiveness to the first lidar range gate, it can be classified as either aerosol, hydrometeor or undefined. The final result is shown in Fig. 3g.*

*The resulting classes are background signal, aerosol, hydrometeor, and undefined. For this algorithm, hydrometeors are defined as cloud (liquid or ice) or precipitation (rain or snow) and do not include aerosol.*

*We also simplified the post-processing section, Sect. 2.3*

*Aerosol is expected to be well-mixed within each aerosol layer, so in order to extract weak aerosol signal and minimize the*
*random noise, $SNR_{co}$ and $SNR_{cross}$ were averaged for 1 hour.*

*As mentioned before, the SNR data in this study have been processed with the background correction algorithm described by Vakkari et al. (2019). Briefly, the noise floor consists of a non-polynomial component, which is obtained from the background checks according to Vakkari et al. (2019) and a polynomial component, which is obtained from a fit to the aerosol- and hydrometeor- free (background) range gates of each $SNR_{co}$ and $SNR_{cross}$ profile (Manninen et al., 2016; Vakkari et al.,*
*2019). Typically, the linear part of the noise floor is much larger than the 2nd order polynomial component, but for extended averaging (more than 1 hour) it is essential to include in the background correction. An example of this is shown in Fig. 4a demonstrating how this 2nd order polynomial component can greatly affect the $\delta$ of aerosol retrieval in aerosol layers with low SNR (Fig. 4d). Previously (Vakkari et al., 2021; Bohlmann et al., 2021), the 2nd order component of the noise floor has been fitted to aerosol- and hydrometeor- free range gates of the SNR profiles based on visual inspection of individual profiles.*
*However, given the large number of profiles analysed in this study, this approach is not feasible and thus we have automated the fitting of the 2nd order polynomial. The fitting algorithm is described in detail in the Sect. S2, and the resulting $SNR_{co}$ and $SNR_{cross}$ profiles are shown in Fig. 4b, 4c.*

*The attenuated backscatter is calculated from the background-corrected $SNR_{co}$. Next, aerosol layer(s) are identified using the Aerosol Identification algorithm. Finally, following Vakkari et al. (2021), the bleed-through corrected $\delta$ in aerosol regions is*
*calculated as*

$$\delta = \frac{SNR_{cross} - B \cdot SNR_{co}}{SNR_{co}} \tag{3}$$

*where B is the estimated bleed-through of each instrument (Table 3).*

*The resulting $\delta$ is shown in Fig. 4d, and the estimation of its uncertainty is presented in Sect. S2. Additionally, the post-processed $\delta$ of aerosol was collected from the whole dataset and compared with the original $\delta$. The result described in Sect. S2 shows*
*that the post-processing procedure substantially improved the $\delta$ of aerosol with low SNR values.*

Line 166: "... 2D Kernel manipulation...". What is this 2D-Kernel manipulation? Maybe a description about the basic idea in a sentence along with a reference or link would be helpful. This way the reader can understand that this method is bout image processing.

Thank you for the suggestion, we have now added more clarifications:

*The Aerosol Identification algorithm developed here utilizes 2D-kernel manipulation, which is a commonly used approach in image processing (Guo et al., 2022; Li et al., 2013; Perreault and Hébert, 2007), to extract various features from the data and to determine the correct class for each data point. A kernel, also referred to as a filter, template, window or mask (Gonzalez and Woods, 2007) is a small 2-D data array. Mathematical operations, such as median, maximum, Gaussian etc., on all values inside the kernel determine its center value. The kernel is run through each data point one by one, replacing its center value*

*with mathematical operations of the neighboring values.*

Lines 299 – 300: "...This is due to the Streamline Pro models were configured to utilize only half of the bandwidth, i.e half the Nyquist velocity...". Could the authors discuss a bit further how the half of bandwidth can affect the noise floor levels? Maybe you could add an extra explanatory sentence along with a reference.

Assuming the noise is thermal noise and is evenly distributed across the frequency spectrum, the noise power follows Johnson-Nyquist noise power equation (Nyquist, 1928; Johnson, 1928):

$$N = kTB,$$

where k is the Boltzmann constant, T is the temperature, and B is the bandwidth. So, reducing the bandwidth by half will reduce the noise power by half. The half bandwidth of the Streamline Pro models were configured by the manufacturer.

Line 307: "...Figure 6 displays the time series of $\delta$ at liquid cloud base for each instrument in the network." Here the authors could also add that they are using the time series of $\delta$ in order to calculate the bleed through (B) and its uncertainty as the mean value and std of $\delta$ over time (as you state in lines 318–320).

*Figure 6 displays the time series and the distribution of $\delta$ at liquid cloud base for each instrument in the network. The time series is used assess the stability of the internal polarizer over time and the distribution is used to calculate the bleed through (B) and its uncertainty. Overall, there is no significant trend in the bleed through of all the instruments.*

Figure 6: Data from the same stations are used in Figures 5 and 6. A suggestion that could help the reader understand faster
Figure 6 is to use the same colors for each station of Figure 5 in Figure 6, too.

We agree with your comment, and have modified the plot accordingly

[Figure]

**Figure 6.** *Time series (left panels) and histogram (right panels) of depolarization ratio (δ) at liquid cloud base in a) Utö-32, b) Utö-32XR, c) Hyytiälä-33, d) Hyytiälä-46, e) Vehmasmäki-53, f) Sodankylä-54. The best estimates of mean (μ) and standard deviation σ of δ at liquid lcoud base in each site are shown in the right panel.*

Section 3.4.1: The effect of relative humidity. In this section the authors mention the diurnal pattern of $\delta$_aerosol and RH below 300 m a.g.l.. How these RH patterns obtained if not measured? Please clarify this also in the caption of Figure 10.

We add clarification in the text

*However, as RH profiles are not measured, we investigate the connection between the surface RH at 2 m a.g.l and the $\delta$ of aerosol below 300 m a.g.l*

We modified the caption

*Diurnal pattern of the particle depolarization ratio of aerosol ($\delta$aerosol) from 90m to 300m a.g.l and relative humidity (RH) at 2m a.g.l respectively...*

**References**

Baars, H., Kanitz, T., Engelmann, R., Althausen, D., Heese, B., Komppula, M., Preißler, J., Tesche, M., Ansmann, A., Wandinger, U., Lim, J. H., Young Ahn, J., Stachlewska, I. S., Amiridis, V., Marinou, E., Seifert, P., Hofer, J., Skupin, A., Schneider, F., Bohlmann, S., Foth, A., Bley, S., Pfüller, A., Giannakaki, E., Lihavainen, H., Viisanen, Y., Kumar Hooda, R., Pereira, S. N., Bortoli, D., Wagner, F., Mattis, I., Janicka, L., Markowicz, K. M., Achtert, P., Artaxo, P., Pauliquevis, T., Souza, R. A., Prakesh Sharma, V., Gideon Van Zyl, P., Paul Beukes, J., Sun, J., Rohwer, E. G., Deng, R., Mamouri, R. E., and Zamorano, F.: An overview of the first decade of PollyNET: An emerging network of automated Raman-polarization lidars for continuous aerosol profiling, Atmospheric Chemistry and Physics, 16, 5111–5137, https://doi.org/10.5194/acp-16-5111-2016, 2016.

Bohlmann, S., Shang, X., Vakkari, V., Giannakaki, E., Leskinen, A., Lehtinen, K. E., Pätsi, S., and Komppula, M.: Lidar depolarization ratio of atmospheric pollen at multiple wavelengths, Atmospheric Chemistry and Physics, 21, https://doi.org/10.5194/acp-21-7083-2021, 2021.

Engelmann, R., Kanitz, T., Baars, H., Heese, B., Althausen, D., Skupin, A., Wandinger, U., Komppula, M., Stachlewska, I. S., Amiridis, V., Marinou, E., Mattis, I., Linné, H., and Ansmann, A.: The automated multiwavelength Raman polarization and water-vapor lidar PollyXT: The neXT generation, Atmospheric Measurement Techniques, 9, 1767–1784, https://doi.org/10.5194/amt-9-1767-2016, 2016.

Gonzalez, R. C. and Woods, R. E.: Digital Image Processing (3rd Edition), publication Title: Prentice-Hall, Inc. Upper Saddle River, NJ, USA ©2006, 2007.

Guo, S., Wang, G., Han, L., Song, X., and Yang, W.: COVID-19 CT image denoising algorithm based on adaptive threshold and optimized weighted median filter, Biomedical Signal Processing and Control, 75, https://doi.org/10.1016/j.bspc.2022.103552, 2022.

Johnson, J. B.: Thermal Agitation of Electricity in Conductors, Physical Review, 32, 97–109, https://doi.org/10.1103/PhysRev.32.97, publisher: American Physical Society, 1928.

Li, S., Kang, X., and Hu, J.: Image fusion with guided filtering, IEEE Transactions on Image Processing, 22, https://doi.org/10.1109/TIP.2013.2244222, 2013.

Manninen, A. J., O'Connor, E. J., Vakkari, V., and Petäjä, T.: A generalised background correction algorithm for a Halo Doppler lidar and its application to data from Finland, Atmospheric Measurement Techniques, 9, 817–827, https://doi.org/10.5194/amt-9-817-2016, 2016.

Murayama, T., Sugimoto, N., Uno, I., Kinoshita, K., Aoki, K., Hagiwara, N., Liu, Z., Matsui, I., Sakai, T., Shibata, T., Arao, K., Sohn, B. J., Won, J. G., Yoon, S. C., Li, T., Zhou, J., Hu, H., Abo, M., Iokibe, K., Koga, R., and Iwasaka, Y.: Ground-based network observation of Asian dust events of April 1998 in east Asia, Journal of Geophysical Research Atmospheres, 106, https://doi.org/10.1029/2000JD900554, 2001.

Nyquist, H.: Thermal Agitation of Electric Charge in Conductors, Physical Review, 32, 110–113, https://doi.org/10.1103/PhysRev.32.110, publisher: American Physical Society, 1928.

Perreault, S. and Hébert, P.: Median filtering in constant time, IEEE Transactions on Image Processing, 16, https://doi.org/10.1109/TIP.2007.902329, 2007.

Sassen, K.: Indirect climate forcing over the western US from Asian dust storms, Geophysical Research Letters, 29, https://doi.org/10.1029/2001gl014051, 2002.

Vakkari, V., Manninen, A. J., O'Connor, E. J., Schween, J. H., Van Zyl, P. G., and Marinou, E.: A novel post-processing algorithm for Halo Doppler lidars, Atmospheric Measurement Techniques, 12, 839–852, https://doi.org/10.5194/amt-12-839-2019, 2019.

Vakkari, V., Baars, H., Bohlmann, S., Bühl, J., Komppula, M., Mamouri, R. E., and O'connor, E. J.: Aerosol particle depolarization ratio at 1565 nm measured with a Halo Doppler lidar, Atmospheric Chemistry and Physics, 21, https://doi.org/10.5194/acp-21-5807-2021, 2021.